# The impacts of modelling prescribed vs. dynamic land cover in a high $CO_2$ future scenario – greening of the Arctic and Amazonian dieback

Sian Kou-Giesbrecht[1], Vivek K. Arora[2], Christian Seiler[3], and Libo Wang[4]

[1]Department of Earth and Environmental Sciences, Dalhousie University, Halifax, NS, Canada
[2]Canadian Centre for Climate Modelling and Analysis, Climate Research Division, Environment Canada, Victoria, BC, Canada
[3]School of Environmental Studies, Queen's University, Kingston, ON, Canada
[4]Climate Processes Section, Climate Research Division, Environment and Climate Change Canada, Toronto, ON, Canada

*Correspondence to*: Sian Kou-Giesbrecht (sian.kougiesbrecht@dal.ca)

**Abstract.** Terrestrial biosphere models are a key tool in investigating the role played by the land surface in the global climate system. However, few models simulate the geographic distribution of biomes dynamically, opting to prescribe them instead using remote sensing products. While prescribing land cover still allows for the simulation of the impacts of climate change on vegetation growth as well as the impacts of land use change, it prevents the simulation of climate change-driven biome shifts, with implications for projecting the future terrestrial carbon sink. Here, we isolate the impacts of prescribed vs. dynamic land cover implementations in a terrestrial biosphere model. We first introduce a new framework for evaluating dynamic land cover (i.e., the spatial distribution of plant functional types across the land surface), which can be applied across terrestrial biosphere models alongside standard benchmarking of energy, water, and carbon cycle variables in model intercomparison projects. After validating simulated land cover, we then show that the simulated terrestrial carbon sink differs significantly between simulations with dynamic vs. prescribed land cover for a high $CO_2$ future scenario. This is because of important range shifts that are only simulated when dynamic land cover is implemented: tree expansion into the Arctic and Amazonian transition from forest to grassland. In particular, the projected change in net land-atmosphere $CO_2$ flux at the end of the 21st century is twice as large in simulations with dynamic land cover than in simulations with prescribed land cover. Our results illustrate the importance of climate change-driven biome shifts for projecting the future terrestrial carbon sink.

## 1 Introduction

Terrestrial biosphere models simulate the exchange of $CO_2$, water, and energy between the atmosphere and the land surface under both current and future climatic and socioeconomic conditions. The results from these models have led to the critical understanding that the terrestrial biosphere currently sequesters approximately a third of anthropogenic $CO_2$ emissions (Friedlingstein et al., 2022). Vegetation currently covers ~75% of global land area and plays a major role in regulating the

land-atmosphere exchange of $CO_2$ (Fisher and Koven, 2020). Vegetation is a dynamic component of the global climate system that responds and feeds back to changes in environmental conditions at timescales ranging from seconds to centuries. Stomata, pores through which plants exchange $CO_2$ and water with the outside air, respond to changes in environmental conditions within seconds (Vialet-Chabrand et al., 2017). Over the course of a day, vegetation responds to diurnal cycles of

energy. Seasonality determines annual cycles of plant phenology and interannual climate variability regulates the land-atmosphere exchange of $CO_2$ over decades. Over decades to centuries, the geographical distribution of natural vegetation can also change as the geographical distribution of different biomes shift with long-term climate variations (Pecl et al., 2017). These changes to vegetation have critical implications for the terrestrial carbon (C) sink as well as feedbacks to the climate system. Because changes to the geographical distribution of natural vegetation occur over such long timescales, terrestrial

biosphere models generally opt to prescribe the geographical distribution of natural vegetation rather than simulate it dynamically. This avoids the challenges associated with accurately reproducing the geographical distribution of natural vegetation which introduces another degree of uncertainty. However, when modelling the terrestrial biosphere over longer timescales, changes to the geographical distribution of natural vegetation are magnified and become critically important (Renwick and Rocca, 2015).

Terrestrial biosphere models generally represent vegetation using a set of plant functional types (PFTs) due to the intractability associated with modelling ecosystem processes for each plant species individually. This classification scheme allows terrestrial biosphere models to simplify the diversity in physiology and abiotic interactions across plant species by clustering plant species by their fundamental structure and function (Box, 1996; Reich et al., 2003). Models may choose to represent a set of PFTs that are distinguished based on leaf form (needleleaf or broadleaf), leaf phenology (evergreen or

deciduous), stature (trees, grasses, or shrubs), photosynthetic pathway ($C_3$ or $C_4$), or geographical location (tropical, temperate, or boreal). When a terrestrial biosphere model prescribes land cover rather than represents it dynamically, the spatial distribution of PFTs across the land surface is specified using current observations of land cover based on remote sensing products (National Research Council, 2008). Remote sensing products are first reclassified into the PFTs that a given model represents (Hartley et al., 2017). Then historical information on land use change is incorporated: crop and pasture

areas have increased while natural vegetation area has correspondingly decreased over the past century (Chini et al., 2021). The resulting land cover forcing allows for the simulation of how deforestation and the conversion of natural vegetation to crop and pasture, i.e., land use change, impacts net terrestrial C sequestration (Houghton et al., 2012). Because the expansion of crop and pasture areas has been greater than climate-driven changes to the geographical distribution of natural vegetation over the historical period, prescribing land cover using a forcing that only accounts for land use change is a reasonable

assumption for simulations over the historical period (although historical climate-driven changes to the geographical distribution of natural vegetation are not negligible (Pecl et al., 2017)). However, prescribing land cover does not capture the shifting ranges of different biomes driven by climate change and its subsequent impacts on the net land-atmosphere $CO_2$ flux. Changes to the geographical distribution of natural vegetation should be especially important for high $CO_2$ future

scenarios with correspondingly strong climate warming because these scenarios exhibit the largest changes to vegetation
productivity (Arora et al., 2020; Koven et al., 2022) and thus the largest potential changes to the ranges of different biomes.

In its simplest form, the dynamic behaviour of vegetation can be classified into two aspects: vertical and horizontal. Vertical changes in vegetation structure include changes to leaf area index, vegetation height, rooting depth, etc. given a certain spatial extent. Horizontal changes in vegetation structure describe changes in this spatial extent. Global change influences both aspects of the dynamic behaviour of vegetation: the response of plant growth to global change drivers such as $CO_2$
fertilisation and climate variability (Dusenge et al., 2019; Huang et al., 2018; Wu et al., 2011), as well as range shifts of different PFTs. The range of a given PFT is modulated by competitive interactions for space and resources with other PFTs and by the bioclimatic limits within which a PFT can exist. Both competitive interactions and bioclimatic limits are affected by climate change (Lenoir and Svenning, 2015; Thomas, 2010; Walther, 2010). While a terrestrial biosphere model with prescribed land cover can capture how global change modulates plant growth of a given PFT within its specified range
(vertical changes), it cannot capture accompanying climate change-driven range shifts of its PFTs (horizontal changes) aside from specified land use change. Conversely, a terrestrial biosphere model with dynamic land cover can capture both variation in plant growth (vertical changes) as well as range shifts (horizontal changes) driven by climate change alongside specified land use change.

The Global Carbon Project produces an annual quantification of the global C budget, which includes an estimate of the
current terrestrial C sink (i.e., the positive global atmosphere-land $CO_2$ flux) from an ensemble of terrestrial biosphere models (Friedlingstein et al., 2022). In the most recent budget, only 3 out of the 16 contributing terrestrial biosphere models implemented dynamic land cover. Furthermore, only 3 out of the 11 contributing Earth System Models to the Coupled Climate–Carbon Cycle Model Intercomparison Project ($C^4MIP$) within the 6th phase of the Coupled Model Intercomparison Project (CMIP), which coordinates the analysis of C cycle interactions, included dynamic land cover (Arora et al., 2020).
Here, we isolate the impacts of prescribed vs. dynamic land cover by comparing two versions of the Canadian Land Surface Scheme Including Biogeochemical Cycles (CLASSIC): a version with prescribed land cover and a version with dynamic land cover. All other aspects of the model are identical between the two versions, including land use change, thereby isolating the effects of land cover implementation (prescribed vs. dynamic). CLASSIC has contributed to the Global Carbon Project since 2016 and CLASSIC is the land component of the Canadian Earth System Model (CanESM), which contributes
to CMIP and the assessment reports of the Intergovernmental Panel on Climate Change (IPCC).

We first demonstrate that CLASSIC with dynamic land cover successfully reproduces the current geographical distribution of natural vegetation when compared to remote sensing products following a statistical framework, which we adapt for the first time to include an evaluation of the geographical distribution of natural vegetation. This framework can be applied across terrestrial biosphere models alongside standard benchmarking of energy, water, and carbon cycle variables. We then
analyse CLASSIC simulations of Shared Socioeconomic Pathway 585 (SSP5-8.5), which describes a "fossil-fueled development scenario" from 2015 to 2100, and examine how climate change could influence the future geographical

distribution of natural vegetation. We identify differences between projections with prescribed land cover vs. dynamic land cover and examine how different land cover implementations impact the future terrestrial C sink.

## 2 Methods

**2.1 CLASSIC overview**

The Canadian Land Surface Scheme Including Biogeochemical Cycles (CLASSIC) (Melton et al., 2020; Seiler et al., 2021) is the successor to and is based on the coupled Canadian Land Surface Scheme (CLASS (Verseghy, 1991; Verseghy et al., 1993)) and the Canadian Terrestrial Ecosystem MODEL (CTEM (Arora and Boer, 2005a; Melton and Arora, 2016)). Older versions of CLASSIC (under the name CLASS-CTEM) have served as the land component in the family of Canadian Earth

System Models (CanESM), including CanESM5 which contributes to CMIP (Swart et al., 2019).

The physical component of CLASSIC simulates fluxes of energy, momentum, and water (Verseghy, 1991; Verseghy et al., 1993). The structural attributes of vegetation are characterized by leaf area index (LAI), canopy mass, vegetation height, and rooting depth, all of which are dynamically simulated by the biogeochemical component of CLASSIC (described below). The soil profile is represented by 20 soil layers, starting with 10 soil layers of 0.1 m thickness followed by soil layers of

increasing thickness up to a soil layer of 30 m thickness for a total depth of 61.4 m. The depth of permeable soil layers and thus the depth to bedrock soil layers varies geographically and is specified based on the SoilGrids250m data set (Hengl et al., 2017). Soil temperature and soil moisture content (liquid and frozen) are simulated for each permeable soil layer. Where the climate permits snow to exist, the temperature, mass, density, and albedo of a single snowpack layer are simulated. The physical calculations yield net radiation, soil heat flux, latent and sensible heat fluxes, evapotranspiration, and runoff at the

land-atmosphere boundary. Each grid cell is simulated independently and there are no lateral transfers of energy or matter between grid cells.

The biogeochemical component of CLASSIC simulates the land-atmosphere exchange of $CO_2$ via photosynthesis, autotrophic respiration, heterotrophic respiration, land use change, and fire (Arora and Boer, 2005a). CLASSIC prognostically simulates the amount of C in vegetation, litter, and soil organic matter pools for each PFT and over the bare

soil fraction in each grid cell. Vegetation C is represented by leaf, stem, and root components, each of which consists of structural and non-structural carbohydrate pools. Photosynthesis generates non-structural carbohydrates which are allocated between the non-structural leaf, stem, and root C pools. Autotrophic respiration occurs from the non-structural leaf, stem, and root C pools (Arora and Boer, 2005a). Non-structural C is converted to structural C within each vegetation component (Asaadi et al., 2018). Leaf, stem, and root turnover transfer C from the vegetation C pool to the litter C pool. In addition to

normal leaf turnover, leaf turnover also occurs due to drought stress, cold stress, and shorter day lengths, affecting leaf phenology (Arora and Boer, 2005a). Land use change transfers C from the vegetation C pool to land use change product C pools (with turnover times corresponding to pulp/paper products and wood products), whereas fire emits C to the atmosphere and also transfers C from the vegetation C pool to the litter C pool (Arora and Boer, 2005b; Arora and Melton, 2018).

Decomposition transfers C from the litter C pool to the soil C pool. Finally, heterotrophic respiration occurs from both the
litter and soil C pools (Melton et al., 2015). While CLASSIC does include a representation of nitrogen cycling (Asaadi and Arora, 2021; Kou-Giesbrecht and Arora, 2022), the interactions between C and nitrogen cycling are not considered in this study.

**2.2 CLASSIC land cover implementation**

Biogeochemical processes in CLASSIC are modelled for nine plant functional types (PFTs): needleleaf evergreen trees
(NE), needleleaf deciduous trees (ND), broadleaf evergreen trees (BE), deciduous broadleaf cold trees (DBC), deciduous broadleaf dry trees (DBD), $C_3$ crops, $C_4$ crops, $C_3$ grasses (C3G), and $C_4$ grasses (C4G). These nine PFTs map directly to four PFTs used for simulating physical processes (needleleaf trees, broadleaf trees, crops, and grasses). When prescribed land cover is implemented, the time-varying fractional coverage of each PFT in each grid cell is specified by a land use forcing (described below). Land use change is the only driver of variation in the fractional coverage of PFTs over time.
When crop area increases, natural vegetation area proportionally decreases. When crop area decreases, natural vegetation area proportionally increases. When dynamic land cover is implemented, the time-varying fractional coverages of $C_3$ crops and $C_4$ crops in each grid cell are specified by a land use forcing and the fractional coverages of natural PFTs in each grid cell evolve due to competition and mortality within the non-crop area of each grid cell. For both prescribed and dynamic land cover implementations, the fractional coverages of all PFTs and bare ground in a grid cell sum to 1:

$$\sum_{n=1}^{N+1} f_n = 1 \qquad (1)$$

$f_n$ is the fractional coverage of a given natural PFT $n \in \{1, ..., N\}$ and $f_{N+1}$ is the fractional coverage of bare ground in a given grid cell. Additionally, a land surface fraction for each grid cell is specified to exclude water bodies (oceans and lakes), glaciers, and ice sheets.

When dynamic land cover is implemented, the fractional coverage of a natural PFT is the result of colonization and mortality
and is described in detail in Melton & Arora (2016) and in the Supplementary Information. Competition between natural PFTs is based on modified Lotka-Volterra equations which describe the interactions between two populations using first-order nonlinear differential equations. In this case, competition is for area in a grid cell rather than for population size. The colonization rate of natural PFT $n$ ($c_n$; day$^{-1}$) is determined by its net primary productivity and leaf area index (described in detail in Appendix A). The mortality rate of natural PFT $n$ ($m_n$; day$^{-1}$) is the sum of intrinsic or age-related mortality,
mortality due to reduced growth, mortality due to fire, and mortality when a PFT exists outside its bioclimatic limits. Intrinsic or age-related mortality is determined by its maximum age (described in detail in Appendix A). Mortality due to reduced growth is determined by its growth rate over the previous year. Mortality due to fire is described in detail in Arora & Melton (2018). Area burned, which generates bare ground, depends on a PFT-specific fire spread rate (where grasses have a higher fire spread rate than trees, and needleleaf trees have a higher spread rate than broadleaf trees), wind speed, and soil
moisture as well as the probability of fire occurrence (which depends on the availability of vegetation biomass as a fuel

source, the combustibility of the fuel source based on soil moisture, and the presence of an ignition source based on lightning and population density) and fire suppression (which depends on population density).

Mortality when a PFT exists outside its bioclimatic limits ($m_{bioclim,n}$; day$^{-1}$) ensures that a given natural PFT $n$ does not venture outside of its bioclimatic envelope. Six bioclimatic indices are used to determine the spatial range of PFTs, representing the physiological limits to their survival: air temperature of the coldest month, air temperature of the warmest month, aridity index (ratio of potential evaporation to precipitation), growing degree days (cumulative number of days with air temperature above 5°C), dry season length (number of consecutive months with precipitation less than potential evaporation), and precipitation surplus (difference between precipitation and potential evaporation). Each bioclimatic index for each grid cell is updated annually on a 25-year timescale using exponential smoothing:

$$X(t + 1) = X(t)e^{-1/25} + x(t)\left(1 - e^{-1/25}\right) \tag{2}$$

$x(t)$ represents a bioclimatic index at year $t$ and $X(t)$ represents the smoothed bioclimatic index at year $t$. This accounts for time lags in the response of vegetation to climate change drivers (Wu et al., 2015). At the beginning of each time step, each grid cell is assigned with a small fractional coverage of each PFT (0.001). Whether, the PFT persists is determined by $m_{bioclim,n}$: $m_{bioclim,n} = 0.25$ when any bioclimatic index is outside its corresponding bioclimatic limit for a given PFT and $m_{bioclim,n} = 0$ when all bioclimatic indices are inside their corresponding bioclimatic limits for a given PFT. Bioclimatic limits for each natural PFT are given in Table B1. These were derived by first calculating the bioclimatic indices for each grid cell using the CRUJRA climate dataset averaged over 1900 to 1920 (Harris et al., 2020). Then, the 1% and 99% quantiles of these bioclimatic indices over all grid cells for each land cover type in the Collection 5 MODIS Global Land Cover Type International Geosphere-Biosphere Programme (IGBP) product (Friedl et al., 2010) (time-averaged between 2001-2010) were calculated and associated with a CLASSIC PFT. Note that this assumes that the current biome ranges are in equilibrium with the 1900-1920 climate due to the migration lag that occurs between climatic change and observed differences in established plant species ranges, especially in long-lived plant species such as trees (Corlett and Westcott, 2013). While this approach accounts for slower range shifts due to climatic change, it does not account for rapid range shifts due to fire disturbance, which have influenced range shifts over recent decades (Macander et al., 2022; Wang et al., 2020).

To simulate competition between PFTs, each natural PFT is given a dominance rank ($i$) where PFTs with higher dominance ranks invade PFTs with lower dominance ranks and bare ground. Tree PFTs are given higher dominance ranks than grass PFTs. Within tree PFTs and within grass PFTs, a PFT with a higher $c_n$ is dominant over all PFTs with lower $c_n$. Each PFT is thus ranked from $1, 2, \ldots, i - 1, i, i + 1, \ldots N$ where the PFT with dominance rank 1 is the most dominant. Overall, the change in fractional coverage of a PFT with dominance rank $i$ is due to (1) its colonization into the areas of PFTs with lower dominance ranks ($i + 1, \ldots, N$) and bare ground ($N + 1$), (2) colonization by PFTs with higher dominance ranks ($i, \ldots, i - 1$) into its area, and (3) its mortality:

$$\frac{df_i}{dt} = (c_i f_{i+1} + \cdots + c_i f_N + c_i f_{N+1}) - (c_1 f_i + \cdots + c_{i-1} f_i) - m_i f_i \tag{3}$$

The change in fractional coverage of bare ground is due to mortality of the PFTs and colonization of bare ground by the PFTs:

$$\frac{df_{N+1}}{dt} = \sum_{n=1}^{N} m_n f_n - \sum_{n=1}^{N} c_n f_{N+1} \qquad (4)$$

The conceptual form of Equations (3) and (4) is different from the standard Lotka-Volterra equation and allows coexistence of PFTs (Arora and Boer, 2006).

## 2.3 Simulation descriptions

We use CLASSIC (offline) to simulate the principal aspects of the land surface energy, water, and C cycles at the global scale over the historical period (1851 – 2020) and over the future period (2015 – 2100) for Shared Socioeconomic Pathways 585 (SSP5-8.5) at a spatial resolution of 1˚. Simulations are described in Table 1.

**Table 1. Description of simulations. Simulations analysed (S1-S5) are indicated for the corresponding time period. Corresponding pre-industrial spin-ups and/or historical simulations (to initialise simulations of the future period) are also indicated.**

| Land cover implementation | Protocol | Pre-industrial period | Historical period (1851 – 2020) | Future period / SSP5-8.5 (2015 – 2100) |
|---|---|---|---|---|
| GLC2000 (prescribed) | TRENDY | S1 spin-up | S1 | na |
| ESACCI (prescribed) | TRENDY | S2 spin-up | S2 | na |
| Dynamic | TRENDY | S3 spin-up | S3 | na |
| ESACCI (prescribed) | ISIMIP | S4 spin-up | S4 historical simulation | S4 |
| Dynamic | ISIMIP | S5 spin-up | S5 historical simulation | S5 |

To evaluate dynamic land cover over the historical period (1851 – 2020), we conducted three simulations: two simulations with prescribed land cover (S1 and S2) and a simulation with dynamic land cover (S3). These three historical simulations were initialised from corresponding pre-industrial spin-ups (described below). Simulations with prescribed land cover use land cover forcings specifying the fractional coverage of the nine CLASSIC PFTs that are derived from two remote sensing products: the Global Land Cover 2000 (GLC2000) product (S1) (Bartholomé and Belward, 2005) and the European Space Agency Climate Change Initiative (ESACCI) Version 2 product (S2) (ESA, 2017). The process of generating a land cover forcing specifying the fractional coverage of all PFTs has three steps. First, the present-day fractional coverages of the model's PFTs are obtained from a remote sensing product, which involves reclassifying the land cover classes in the remote sensing product to the model's PFTs. A. Wang et al. (2006) describes how the 22 land cover categories in the GLC2000 product are reclassified to the nine CLASSIC PFTs. L. Wang et al. (2023) show how the 37 land cover categories in the ESACCI product from 2018 are reclassified to the nine CLASSIC PFTs. Second, the fractional coverages of crop PFTs are replaced with values from a land use change forcing for a given year in the present day and the fractional coverages of

natural PFTs are adjusted accordingly such that the total fractional coverage of vegetation is unchanged in that year. Third, the timeseries of fractional coverages of crop PFTs from the land use change forcing is incorporated by adjusting the fractional coverages of natural PFTs over a given time period relative to the single-year land cover forcing generated in the previous step. These three steps are used to generate a land cover forcing specifying the fractional coverage of all model PFTs derived from a remote sensing product over a given time period. Simulations with dynamic land cover use land cover forcings that only specify the fractional coverages of crop PFTs.

S1-S3 follow the TRENDY protocol (used for contributions to the Global Carbon Project) (Friedlingstein et al., 2022). The TRENDY protocol uses the merged monthly Climate Research Unit (CRU) and 6-hourly Japanese 55-year Reanalysis (JRA-55) dataset from Harris et al. (2020), atmospheric $CO_2$ forcing from Dlugokencky and Tans (2022), population density forcing from HYDE3.3 (Klein Goldewijk et al., 2017), and land use change forcing from the LUH2-GCB2022 (Land-Use Harmonization 2) dataset (Chini et al., 2021; Hurtt et al., 2020; Klein Goldewijk et al., 2017). S1-S3 were initialised from corresponding pre-industrial spin ups (Table 1). We conducted different pre-industrial spin ups for each simulation that used the corresponding land cover implementation. Pre-industrial simulations used atmospheric $CO_2$, land cover, and population density forcings corresponding to the year 1851. Pre-industrial simulations used meteorological forcings from 1901 to 1920 repeatedly. We ran the pre-industrial spin up until the C pools came into equilibrium. A threshold of 0.05 Pg C yr$^{-1}$ for the global net atmosphere-land $CO_2$ flux was used to assess if equilibrium was achieved. Note that all simulations exclude Antarctica and Greenland.

To examine prescribed vs. dynamic land cover over the future period (2015 – 2100) for SSP5-8.5 (Riahi et al., 2017), we conducted two simulations: one simulation with prescribed land cover from the ESACCI-derived land cover product (S4) and one simulation with dynamic land cover (S5). These future simulations for SSP5-8.5 were initialised from the end of their corresponding historical simulations, which were in turn initialised from their corresponding pre-industrial spin-ups (described below). S4 and S5 use bias-corrected meteorological forcings from CanESM5. They use atmospheric $CO_2$, land use change, and population density forcings from CMIP and the Inter-Sectoral Impact Model Intercomparison Project (ISIMIP) (Buchner and Reyer, 2021; Warszawski et al., 2014). Our study uses a single meteorological forcing (from the corresponding model CanESM5). The effects of different meteorological forcings have been explored elsewhere (Arora et al., 2023) as have the effects of bias correction (Seiler et al., in prep). S4 and S5 were initialised from the end of their corresponding historical simulations, which were in turn initialised from their corresponding pre-industrial spin-ups (Table 1). We conducted different pre-industrial spin ups for each simulation. Pre-industrial simulations used atmospheric $CO_2$, land use change, and population density forcings corresponding to the year 1851. Pre-industrial simulations used pre-industrial meteorological forcings provided by ISIMIP.

## 2.4 Model evaluation

We evaluated historical simulations using the Automated Model Benchmarking R (AMBER) package developed by Seiler et al. (2021), which quantifies model performance in reproducing observation-based datasets using a skill score system that is based on ILAMB (Collier et al., 2018). Five scores compare model output to a reference dataset for a given variable, assessing the simulated time-mean bias ($S_{bias}$), monthly centralised root-mean-square-error ($S_{rmse}$), seasonality ($S_{phase}$), inter-annual variability ($S_{iav}$), and spatial distribution ($S_{dist}$) in comparison to the observation-based dataset. Scores are

dimensionless and range from 0 to 1, where higher values indicate better model performance. The overall score for each variable ($S_{overall}$) is calculated as:

$$S_{overall} = \text{mean}(S_{bias}, S_{rmse}, S_{phase}, S_{iav}, S_{dist}) \tag{5}$$

For variables with more than one observation-based dataset, we calculated a benchmark score for the observation-based datasets, which quantifies how well the independently derived observation-based datasets agree with each other, thereby

providing an estimate of the uncertainty of the observation-based datasets themselves. The benchmark score was calculated by iteratively comparing pairs of independently derived observation-based datasets, where one observation-based dataset is treated as model output and the other observation-dataset is treated as the reference dataset. If the two observation-based datasets were the same, the benchmark score would be 1. The calculation of each score and benchmark scores is described in detail in Seiler et al. (2022).

We adapted this statistical framework, which operates at a grid cell scale, to evaluate land cover fraction for each natural PFT within a grid cell. For simulations with dynamic land cover, we evaluated the land cover fraction for each natural PFT against both the GLC2000-derived land cover product and the ESACCI-derived land cover product. We compared the GLC2000-derived land cover product and the ESACCI-derived land cover product to obtain a benchmark score that quantifies the uncertainty of these land cover products. Note that a qualitative assessment of dynamic land cover at a spatial

resolution of 2.81˚ in CLASSIC was presented in Melton & Arora (2016).

Additionally, we evaluated the standard set of energy, water, and C cycle variables following Seiler et al. (2022). Specifically, we evaluated aboveground biomass C (AGB), surface albedo (ALBS), area burned (BURNTAREA), soil C (CSOIL), vegetation C (CVEG), fire emissions (FIRE), gross primary productivity (GPP), soil heat flux (HFG), latent heat flux (HFLS), sensible heat flux (HFSS), leaf area index (LAI), runoff (MRRO), soil moisture (MRSLL), net biome

productivity (NBP), net ecosystem exchange (NEE), ecosystem respiration (RECO), net surface long wave radiation (RLS), net surface radiation (RNS), net surface shortwave radiation (RSS), and snow water equivalent (SNW). Observation-based datasets used for evaluation are summarized in Table B2.

## 3 Results

### 3.1 Historical simulations and evaluation

CLASSIC with dynamic land cover successfully reproduces the global area of trees, grasses, and total natural vegetation over the historical period, falling within the range of ESACCI-derived and GLC2000-derived land cover products as shown in Figure 1. Simulated global tree, grass, and natural vegetation areas are closer to the ESACCI-derived land cover product than the GLC2000-derived land cover product for the present day (2000-2020). Over the historical period, simulated global natural vegetation area decreases due to increasing crop area, i.e., land use change (Figure B1). Land use change is the only

driver of variation in natural vegetation area in the ESACCI-derived and GLC2000-derived land cover products. CLASSIC with dynamic land cover simulates a smaller decrease in natural vegetation area than that attributable to land use change. This is because decreasing natural vegetation area due to land use change is partially offset by increasing natural vegetation area driven by stimulated plant growth from $CO_2$ fertilisation and climate change over the historical period.

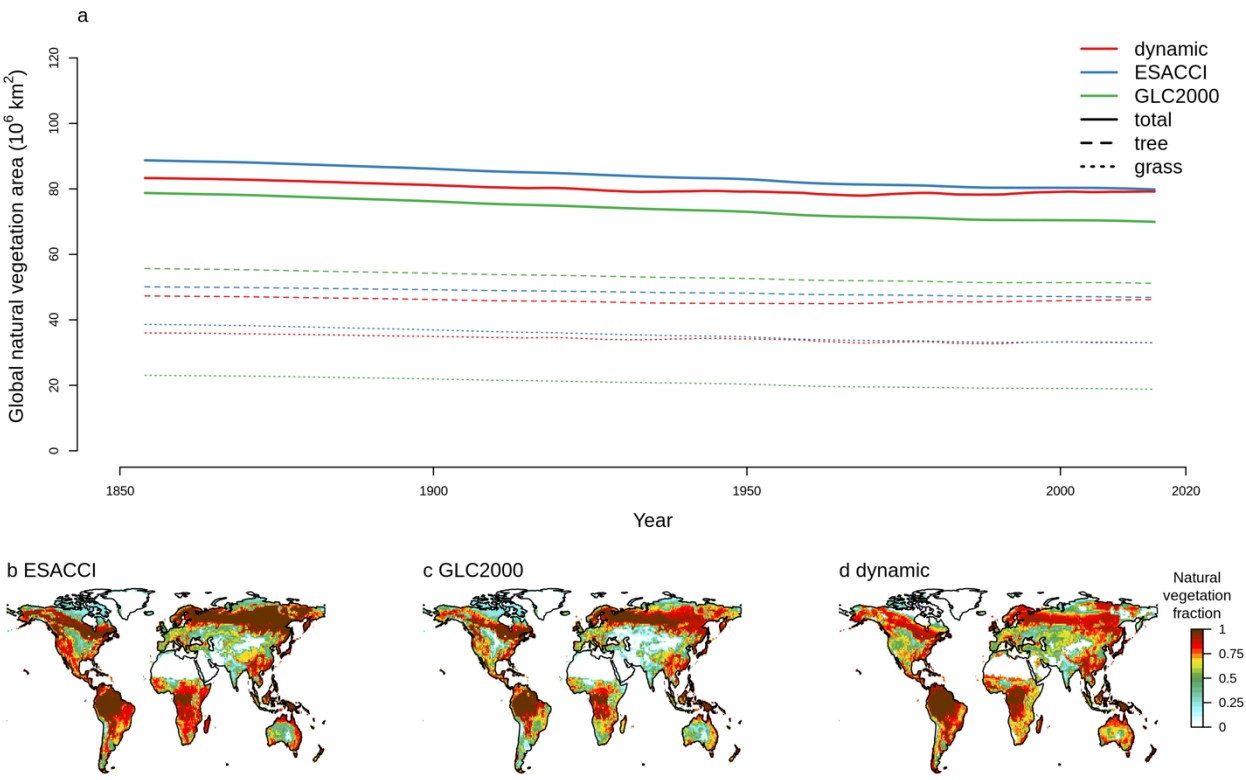

**Figure 1. Natural vegetation area over the historical period. a. Global area of natural vegetation, trees, and grasses simulated by CLASSIC with dynamic land cover in comparison to ESACCI-derived and GLC2000-derived land cover products. Natural vegetation fraction per grid cell in the b. ESACCI-derived land cover product, c. GLC2000-derived land cover product, and d. simulated by CLASSIC with dynamic land cover. Simulations and land cover products are averaged over 2000-2020 in b-d. Natural vegetation includes seven natural PFTs (described in Section 2.2), excluding crop area and bare ground.**

CLASSIC with dynamic land cover also broadly reproduces the latitudinal distribution of the natural PFTs in comparison to ESACCI-derived and GLC2000-derived land cover products as shown in Figure 2. However, there remain some differences between simulations and observations. CLASSIC does not simulate needleleaf evergreen trees between 20°N and 40°N whereas observations suggest that this PFT exists at these latitudes (e.g., high-elevation tropical and subtropical coniferous forests). Rather, CLASSIC simulates broadleaf deciduous cold trees at these latitudes whereas observations suggest that this

PFT declines between 40°N and 20°N. Furthermore, CLASSIC slightly overestimates $C_4$ grass area and slightly underestimates $C_3$ grass area in the Northern Hemisphere. Reasons for these disagreements are discussed below.

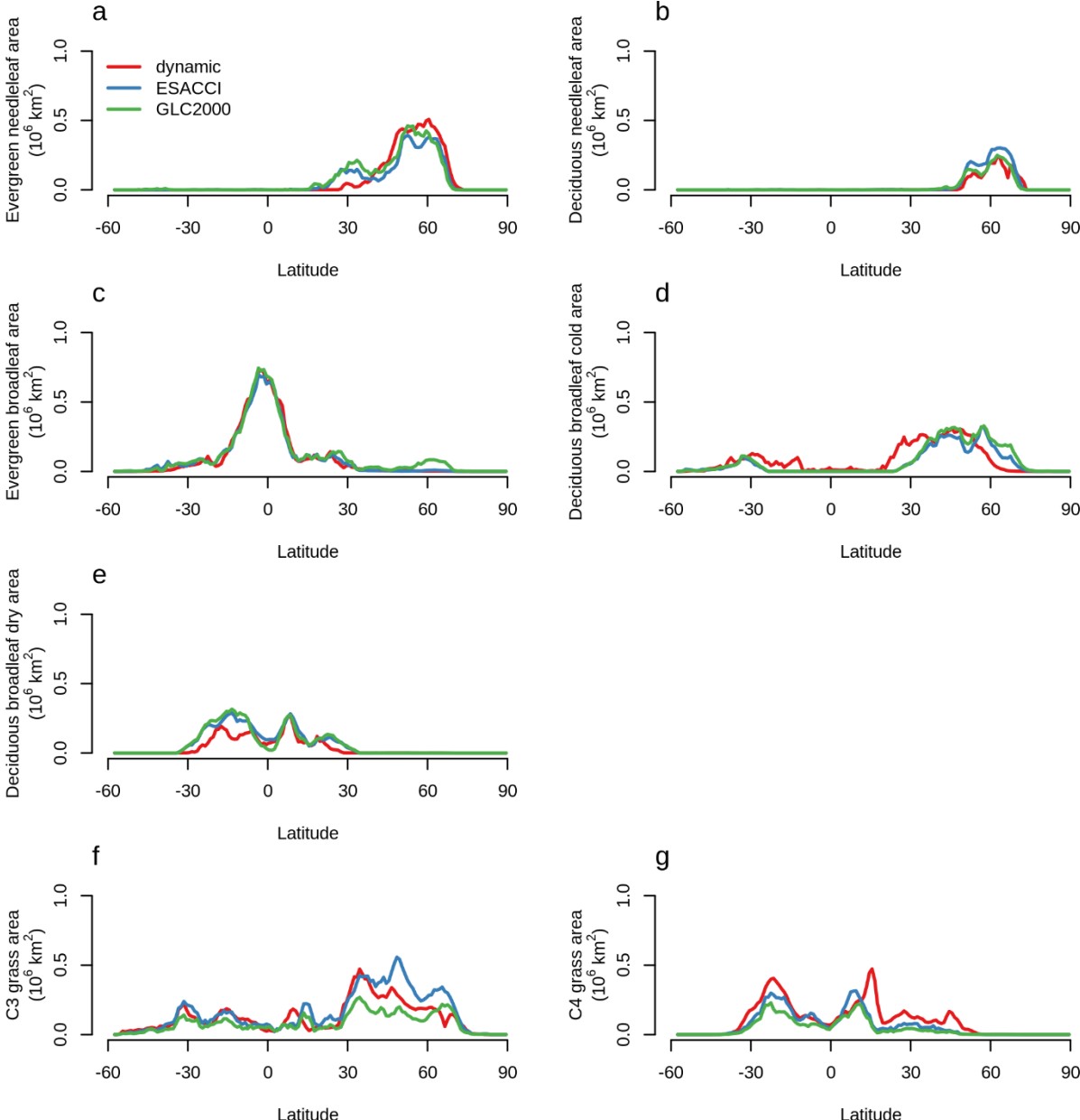

**Figure 2. Latitudinal distributions of each natural plant functional type (PFT) simulated by CLASSIC with dynamic land cover in comparison to ESACCI-derived and GLC2000-derived land cover products. a. Evergreen needleleaf, b. deciduous needleleaf, c. evergreen broadleaf, d. deciduous broadleaf cold, e. deciduous broadleaf dry, f. $C_3$ grass, and g. $C_4$ grass. Simulations and land cover products are averaged over 2000-2020.**

Both CLASSIC with prescribed land cover (simulated with an ESACCI-derived and GLC2000-derived land cover products) and CLASSIC with dynamic land cover reasonably capture the primary C, water, and energy cycle variables (Figure B2). The agreement between CLASSIC simulations and observations is quantified by scores of model performance in

reproducing observation-based datasets for each biogeochemical and biophysical variable. These scores indicate that CLASSIC with dynamic land cover performed similarly to CLASSIC with prescribed land cover (Figure B3). Figures 3a and 3b show model scores (averaged across all CLASSIC simulations, with both prescribed and dynamic land cover) plotted against benchmark scores for biogeochemical and biophysical variables, respectively. Figure 3 shows the correspondence between model performance in reproducing observation-based datasets (i.e., model scores) and the uncertainty of the

observation-based datasets themselves (i.e., benchmark scores). Model scores for both versions of CLASSIC (prescribed and dynamic land cover) and benchmark scores are similar, falling close to the 1:1 line, which indicates that model performance is comparable to the uncertainty of the observation-based datasets themselves. Both versions of CLASSIC perform slightly better for biophysical variables (Figure 3b) than for biogeochemical variables (Figure 3a). Finally, both CLASSIC with prescribed and dynamic land cover simulate the terrestrial C sink over the historical period reasonably well in comparison to

other models in the Global Carbon Project (Friedlingstein et al., 2022) (Figure B4).

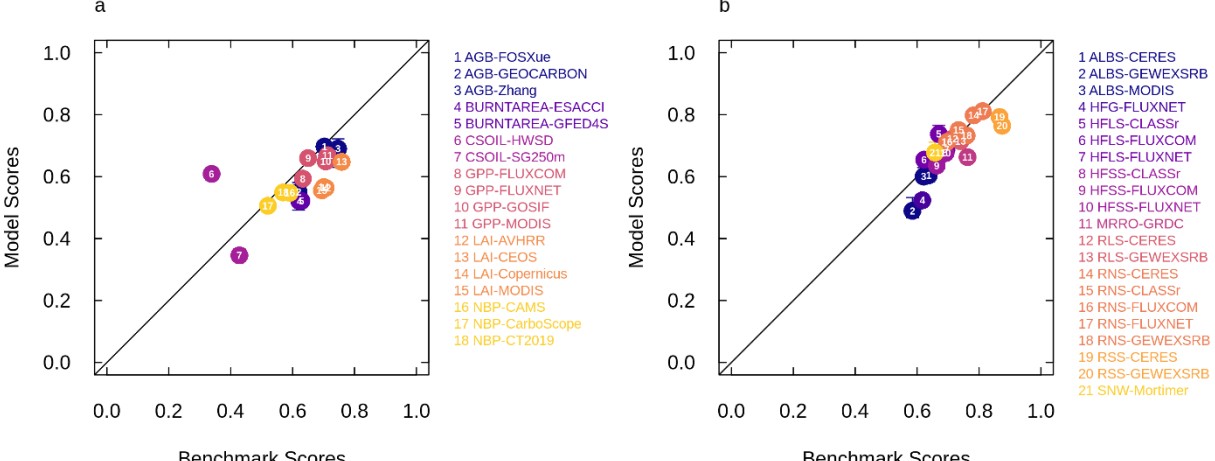

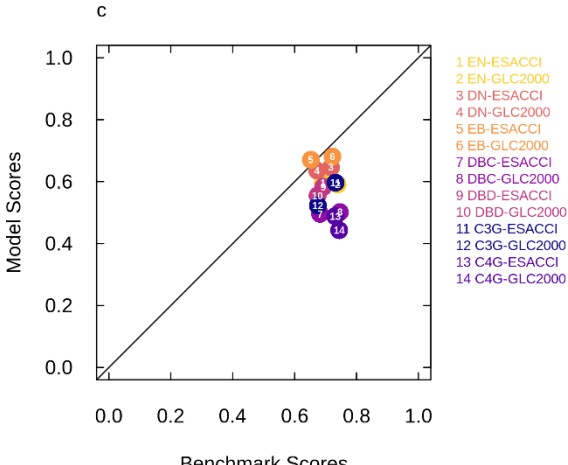

**Figure 3. Model performance in reproducing observation-based datasets (model scores) relative to the uncertainty of the observation-based datasets themselves (benchmark scores) for a. biogeochemical variables, b. biophysical variables, and c. land cover fractions of each natural plant functional type (PFT). Model scores are averaged across simulations of CLASSIC with prescribed land cover (with an ESACCI-derived land cover forcing and a GLC2000-derived land cover forcing) and dynamic land cover for biogeochemical and biophysical variables. Whiskers indicate the maximum and minimum model scores. Abbreviations for C, water, and energy variables and observation-based datasets are described in the Methods. Abbreviations for PFTs are evergreen needleleaf (EN), deciduous needleleaf (DN), evergreen broadleaf (EB), deciduous broadleaf cold (DBC), deciduous broadleaf dry (DBD), $C_3$ grass (C3G), and $C_4$ grass (C4G).**


When dynamic land cover is implemented in CLASSIC, the ability of CLASSIC to reproduce land cover fractions of each natural PFT can be assessed alongside C, water, and energy variables as shown in Figure 3c. Additionally, benchmark scores can be calculated for the land cover fractions of each natural PFT by comparing the ESACCI-derived land cover product to the GLC2000-derived land cover product. Model scores tend to be lower than benchmark scores, especially for tropical PFTs (broadleaf deciduous dry trees and $C_4$ grasses). For these PFTs uncertainty is introduced because remote sensing products do

not distinguish between cold and dry broadleaf deciduous trees and do not distinguish between $C_3$ and $C_4$ grasses. This distinction is made during the reclassification of the land cover categories to the nine CLASSIC PFTs and introduces some subjectivity (Wang et al., 2023b). Specifically, deciduous trees above 30° in both hemispheres are assumed to be cold deciduous, deciduous trees below 20° in both hemispheres are assumed to be dry deciduous, and the fraction of trees that are cold vs. dry deciduous varies linearly between 0 and 1 with latitude for deciduous trees between 20° and 30°. The separation

of grasses into $C_3$ and $C_4$ is based on the time-invariant global product from (Still et al., 2003).

## 3.2 Future simulations

For SSP5-8.5, CLASSIC with dynamic land cover simulates increasing global area of trees, grasses, and total natural vegetation (Figure 4a). Global natural vegetation area decreases in the ESACCI-derived land cover product due to land use change (increasing crop area), which is again the only driver of variation in natural vegetation area in the ESACCI-derived

land cover product (Figure 4a, Figure B1). Decreasing natural vegetation area due to land use change occurs primarily in South America, Africa, and Asia (Figure 4b). CLASSIC with dynamic land cover simulates increasing natural vegetation area because the loss of natural vegetation area due to land use change is offset by increasing natural vegetation area especially at high latitudes (Figure 4c).

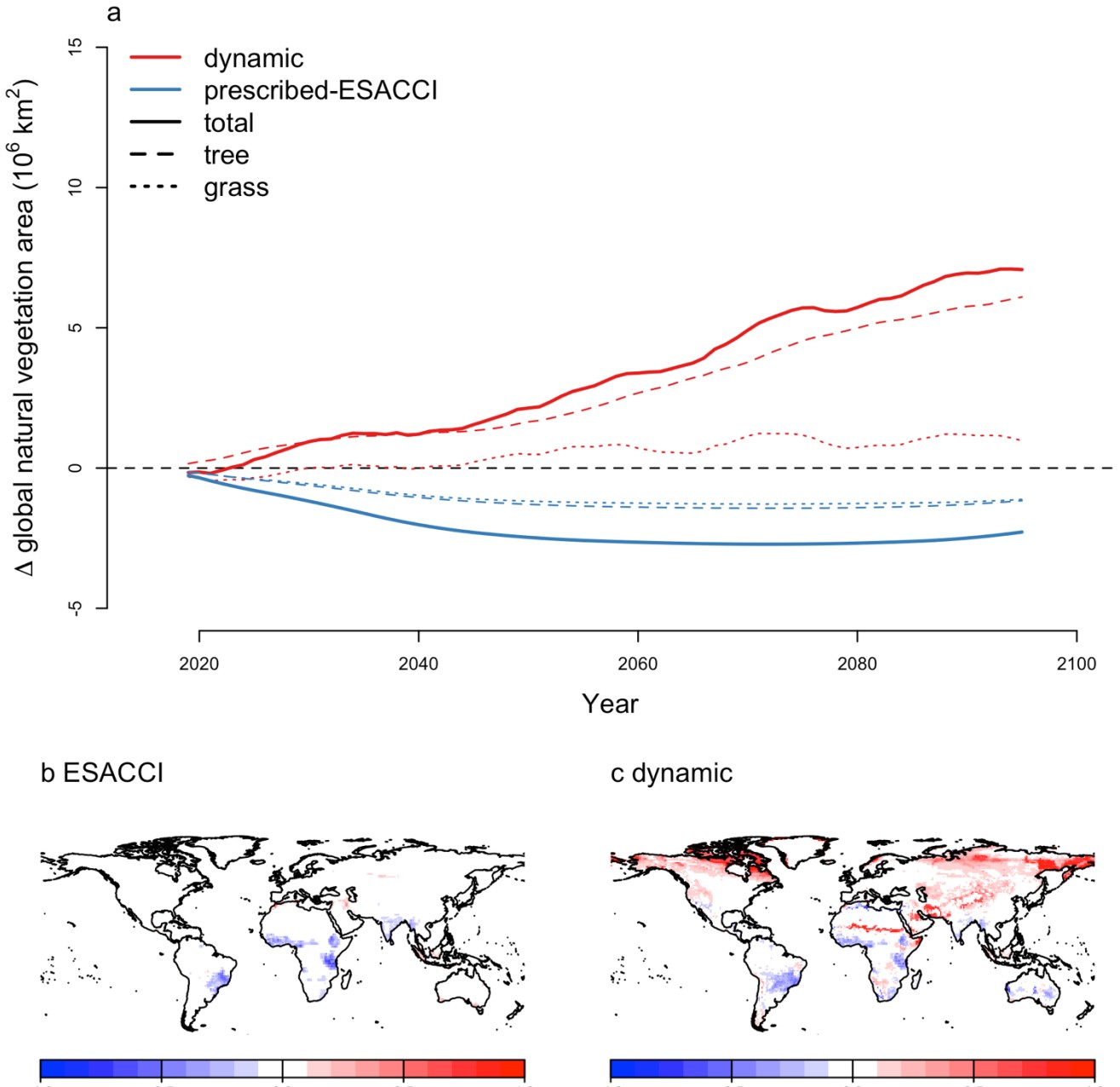

**Figure 4. Change in natural vegetation area for SSP5-8.5 (2015 – 2100). a.** Change in global area of natural vegetation, trees, and grasses simulated by CLASSIC with dynamic land cover in comparison to the ESACCI-derived land cover product. Change in natural vegetation fraction per grid cell **b.** in the ESACCI-derived land cover product and **c.** simulated by CLASSIC with dynamic land cover. Differences reflect the change between the averages calculated over 1995-2015 and 2080-2100. Only grid cells with a significant temporal trend are shown (P < 0.05; assessed with a Mann-Kendall trend test). Figure B5 shows the change in fraction per grid cell for each PFT simulated by CLASSIC with dynamic land cover. Figure B13 shows the absolute value of the natural

**vegetation fraction per grid cell averaged over 1995-2015 and averaged over 2080-2100 simulated by CLASSIC with prescribed land cover and CLASSIC with dynamic land cover.**

At high latitudes, CLASSIC with dynamic land cover simulates increasing areas of needleleaf evergreen trees, broadleaf deciduous cold trees, and $C_3$ and $C_4$ grasses, expanding into what was previously bare ground (Figure B5). This corresponds to increasing temperatures at high latitudes (Figure B6). In particular, $C_4$ grasses, which are currently primarily found at low latitudes, expand into high latitudes due to their dependence on temperature (Luo et al., 2024). At low latitudes, CLASSIC with dynamic land cover simulates decreasing natural vegetation area in South America but variably decreasing and increasing natural vegetation area in Africa and Asia. In the Amazon, the area of broadleaf evergreen trees decreases while the areas of $C_3$ and $C_4$ grasses increase (Figure B5). This corresponds to decreasing precipitation in this region (Figure B6). Conversely, in Africa and Asia, the area of broadleaf evergreen trees increases while the areas of $C_3$ and $C_4$ grasses decrease (Figure B5). This corresponds to increasing precipitation in this region (Figure B6). As an exception, in the Sahel, the areas of $C_3$ and $C_4$ grasses increase, expanding into what was previously bare ground (Figure B5). This also corresponds to increasing precipitation in this region (Figure B6).

CLASSIC with prescribed land cover (simulated with an ESACCI-derived land cover product) and CLASSIC with dynamic land cover both simulate increasing NPP at higher latitudes for SSP5-8.5 (Figure 5a-b). NPP increases to a greater extent at higher latitudes in the simulation with dynamic land cover because of increasing natural vegetation area, especially that of broadleaf trees at the expense of needleleaf trees (Figures 5c-f and 6a-c). NPP changes are similar at lower latitudes in both versions of CLASSIC (Figure 5a-b). Both versions of CLASSIC project decreasing NPP in the Amazon (Figure 5a-b). NPP decreases in simulations with prescribed land cover simulations due to decreasing precipitation in the Amazon (Figure B6). NPP decreases slightly less in the simulation with dynamic land cover because decreasing tree area and NPP are offset by increasing grass area and NPP in the Amazon (Figures 5c-f and 6d-f). In the Sahel, NPP increases slightly in the simulation with dynamic land cover but decreases slightly in the simulation with prescribed land cover (Figure 5a-b). NPP decreases in simulations with prescribed land cover due to land use change in the Sahel (Figure 6h). NPP increases slightly in the simulation with dynamic land cover because of increasing grass area, outweighing the effects of NPP in the Sahel (Figures 5c-f and 6g-i). Both versions of CLASSIC simulate similar increases in vegetation C (Figure B7 and B8). CLASSIC with dynamic land cover simulates a smaller decrease in soil C due to increased natural vegetation area at high latitudes (Figure B7 and B9).

CLASSIC dynamically simulates fire (Arora and Melton, 2018). When prescribed land cover is implemented, fire reduces vegetation biomass density, whereas when dynamic land cover is implemented, fire both reduces vegetation biomass density and creates bare ground that can then be colonized by a different plant functional type. CLASSIC with dynamic land cover simulates slightly higher area burned and fire $CO_2$ emissions than CLASSIC with prescribed land cover at the global scale due to higher natural vegetation area (Figure B10), but spatial patterns are relatively similar between both versions of CLASSIC (Figure B11 and B12). In particular, neither version of CLASSIC simulates substantial changes in high latitude fires which CLASSIC underestimates (Arora and Melton, 2018). CLASSIC with dynamic land cover simulates lower fire in

390  the Amazon and higher fire in the Sahel, corresponding to lower and higher natural vegetation area in the Amazon and Sahel, respectively. This suggests that there are minor differences in fire occurrence between different PFTs which should be improved in future model development.

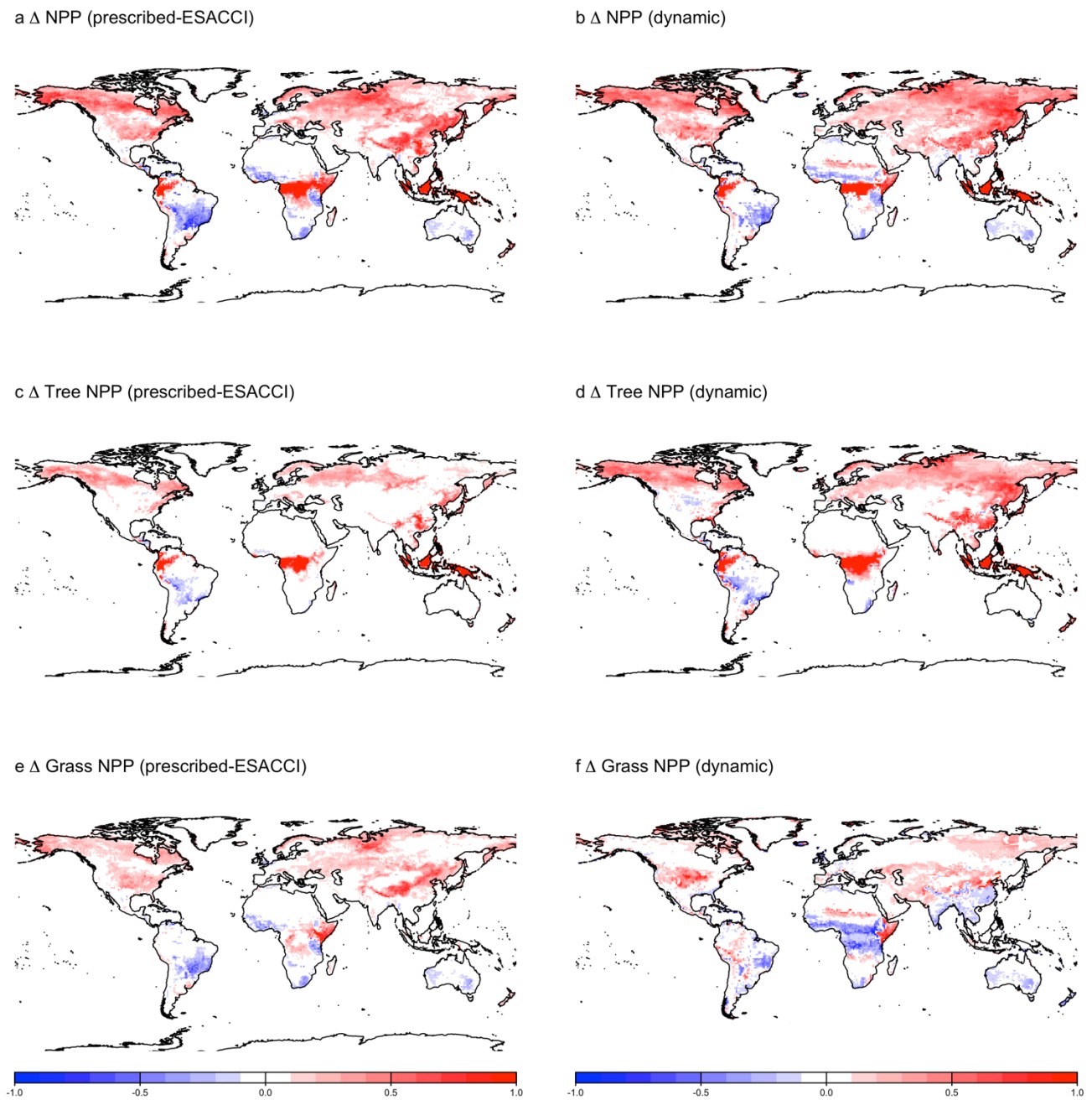

**Figure 5. Change in net primary productivity (NPP) simulated by CLASSIC with prescribed land cover (with an ESACCI-derived**
395  **land cover forcing) and CLASSIC with dynamic land cover for SSP5-8.5 (2015 – 2100). ab. Total NPP, cd. tree NPP, and ef. grass**

NPP. Differences reflect the change between the average over 1995-2015 and 2080-2100. Only grid cells with a significant temporal trend are shown (P < 0.05; assessed with a Mann-Kendall trend test).

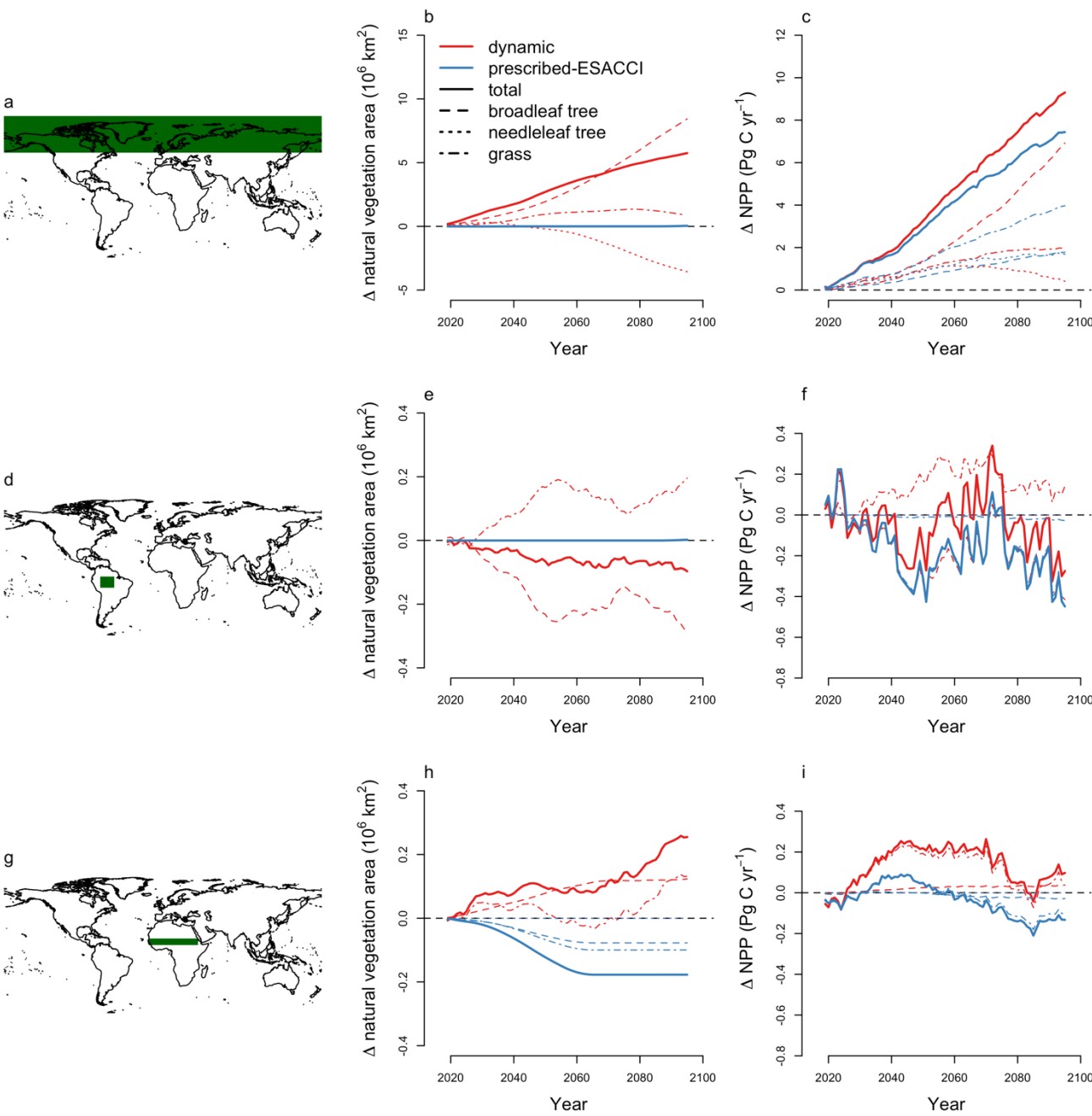

**Figure 6. Change in natural vegetation area and net primary productivity (NPP) at high latitudes, the Amazon, and the Sahel by CLASSIC with prescribed land cover (with an ESACCI-derived land cover forcing) and CLASSIC with dynamic land cover for SSP5-8.5 (2015 – 2100). a. Map of high latitudes. b. Change in global area of natural vegetation, broadleaf trees, needleleaf trees, and grasses in high latitudes. c. Change in NPP in high latitudes. d. Map of the Amazon. e. Change in global area of natural**

**vegetation, broadleaf trees, needleleaf trees, and grasses in the Amazon. f. Change in NPP in the Amazon. d. Map of the Sahel. e. Change in global area of natural vegetation, broadleaf trees, needleleaf trees, and grasses in the Sahel. f. Change in NPP in the Sahel.**

Net biome productivity (NBP), i.e., the net land-atmosphere $CO_2$ flux (which includes photosynthesis, autotrophic and heterotrophic respiration, fire emissions, and land use change emissions), increases in CLASSIC simulations of SSP5-8.5 with both prescribed and dynamic land cover (Figure 7a). The increase in NBP is almost twofold greater in the simulation with dynamic land cover (5.28 Pg C $yr^{-1}$ averaged over 2080 – 2100) than in the simulation with prescribed land cover (2.29 Pg C $yr^{-1}$ averaged over 2080 – 2100). At the global scale, this increase in NBP is driven by increasing NPP (Figure 7b-c) which is primarily due to $CO_2$ fertilisation in both versions of CLASSIC. In CLASSIC with dynamic land cover, there is a stronger NPP increase at high northern latitudes (Figure 7c). NPP increases at low latitudes in both versions of CLASSIC because increasing NPP in Africa and Asia outweighs decreasing NPP in South America. The change in NPP at low latitudes is similar between both versions of CLASSIC despite changing tree vs. grass areas as described above due to offsetting effects. Surface albedo decreases due to snow loss driven by increasing temperature in both versions of CLASSIC (Figure 7d), but this decrease is slightly stronger at high northern latitudes in CLASSIC with dynamic land cover due to the expansion of natural vegetation into higher latitudes (Figure 7e). Evapotranspiration increases in both versions of CLASSIC (Figure 7f), but this increase is slightly stronger at high northern latitudes in CLASSIC with dynamic land cover due to the expansion of natural vegetation into higher latitudes (Figure 7g).

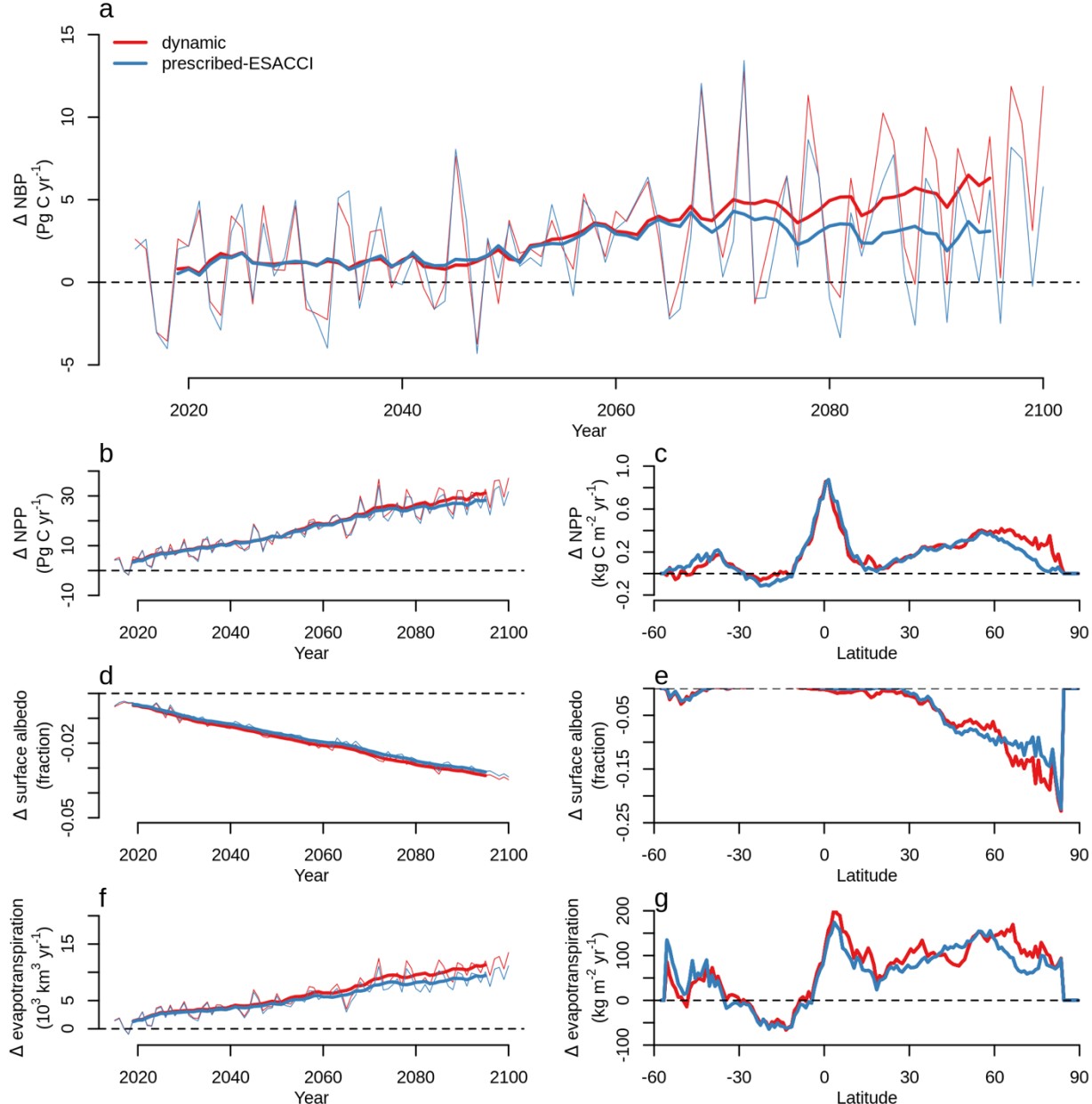

420

**Figure 7. Change in a. net biome productivity (NBP), bc. net primary productivity (NPP), de. surface albedo, and fg. evapotranspiration simulated by CLASSIC with prescribed land cover (with an ESACCI-derived land cover forcing) and CLASSIC with dynamic land cover for SSP5-8.5 (2015 – 2100). For the latitudinal distributions, CLASSIC simulations were averaged over 2080-2100. Thin lines indicate the annual value and thick lines indicate the moving average over 10 years.**

425 **4. Discussion**

Dynamic land cover is rarely implemented in terrestrial biosphere models (e.g., in only 3 out of 11 contributing models to C[4]MIP within the 6[th] phase of CMIP) and the simulated geographical distribution of natural vegetation has not yet been

evaluated robustly as it lacked the methodology to do so. The Global Carbon Project uses a statistical framework to evaluate the ability of its models to reproduce observations of the C, water, and energy cycles – the International Land Model Benchmarking (ILAMB) framework (Collier et al., 2018), which has since been expanded to include a method to quantify the uncertainty of the observation-based datasets themselves (Seiler et al., 2022). We build on this to introduce a framework with which the dynamically simulated geographical distribution of natural vegetation can be quantitatively evaluated against observations. We use this framework to show that CLASSIC with dynamic land cover successfully simulates the geographical distribution of its natural PFTs for the present day, reproducing observations (Figures 1-3). This framework can be applied across terrestrial biosphere models to evaluate dynamic land cover alongside evaluations of the principal C, water, and energy cycle processes.

Future projections for Shared Socioeconomic Pathway 5-8.5 ("fossil-fueled development scenario", SSP5-8.5) differ significantly between simulations with prescribed and dynamic land cover because in the latter the geographical distribution of natural vegetation is able to respond to changing environmental conditions. The net land-atmosphere $CO_2$ flux (or net biome productivity, NBP) is nearly twice as large in the simulation with dynamic land cover (5.28 Pg C $yr^{-1}$ averaged over 2080 – 2100) than the simulation with prescribed land cover (2.29 Pg C $yr^{-1}$ averaged over 2080 – 2100) (Figure 7). This difference is attributed to three important range shifts that are only simulated when dynamic land cover is implemented: (1) the expansion of trees into high latitudes, which increases NBP, (2) the recession of trees accompanied by the expansion of grasses in the Amazon, which decreases NBP, and (3) the expansion of grasses into the Sahel, which increases NBP (Figures 5, 6, and 7). These responses occurred over the historical period as well but to a much smaller extent (Figure 1). We now discuss each of these range shifts in greater detail.

"Greening of the Arctic" describes the expansion of vegetation into high-latitude regions which occurs as previously uninhabitable regions become habitable due to increasing temperatures (Chen et al., 2011; Elmendorf et al., 2012; Myers-Smith et al., 2020; Pearson et al., 2013; Piao et al., 2020; Tape et al., 2006). While terrestrial biosphere models with prescribed land cover capture increasing plant growth at high latitudes within their present-day range due to increasing temperatures (Wang et al., 2023a), these models are not able to simulate range shifts. Our results suggest these range shifts could make a sizeable contribution to the terrestrial C sink via C sequestration in vegetation and soils at high latitudes due to the northward expansion of trees and grasses. Fire alongside rising temperature is known to drive a transition from needleleaf trees to broadleaf trees (Hisano et al., 2021). Because broadleaf trees are less flammable than needleleaf trees this transition could reduce fire activity (Baltzer et al., 2021; Johnstone and Chapin, 2006; Mack et al., 2021). These processes are not represented explicitly here but warrant further study, especially given their importance and significant biases across land surface models at high latitudes (Braghiere et al., 2023; Wang et al., 2021). Although the greening of the Arctic drives increased terrestrial C sequestration at high latitudes, it also drives other important feedbacks to the climate system, such as reduced albedo (Chapin et al., 2005) and enhanced evapotranspiration (Swann et al., 2010).

"Amazonian dieback" describes the combination of deforestation and the increased mortality of trees in the Amazon rainforest due to changing climate in the region (Malhi et al., 2008; Parry et al., 2022). Tree mortality occurs as previously

habitable regions become inhabitable due to the sensitivity of trees to drought and heat stress (Aleixo et al., 2019; Phillips et al., 2009). While the direct effects of reduced precipitation and elevated temperature on plant growth as well as deforestation are captured by terrestrial biosphere models with prescribed land cover, these models are incapable of simulating transitions from forest to grassland. Fire modulates these transitions, as grasses rapidly regenerate post-fire and are more flammable than trees, thereby intensifying fire activity (Davidson et al., 2012). Finally, Amazonian dieback self-amplifies via feedbacks to the climate system, as decreased forest cover decreases evapotranspiration which could amplify regional drought (Zemp et al., 2017). The effects of these feedbacks as well as the feedbacks associated with the greening of the Artic are not evaluated in our offline simulations and warrant further study.

Finally, the "greening of the Sahel" describes the expansion of vegetation into the Sahara which occurs as previously uninhabitable regions become habitable due to increasing precipitation (Brandt et al., 2015; Herrmann et al., 2005; Olsson et al., 2005). This range shift is not captured in terrestrial biosphere models with prescribed land cover. However, there is large uncertainty in future projections of precipitation in this region (Monerie et al., 2020). While less explored than the previous two effects, greening of the Sahel is regionally important and could make a non-trivial contribution to the global C, water, and energy cycles, warranting further study.

These responses are consistent with those of other terrestrial biosphere models that implement dynamic land cover. Both Scholze et al. (2006) and Alo & Wang (2008) simulated increasing tree area at high latitudes and the replacement of trees with grasses in the Amazon. Sitch et al. (2008), which compared four models with dynamic land cover, found similar results, although the magnitudes of the responses differed between models. Port et al. (2012) conducted and examined a fully coupled simulation with a land model with dynamic land cover and similarly observed both greening of the Arctic and Amazonian dieback. Because there is large uncertainty in climate projections in the Sahel region, greening of the Sahel is often but not always simulated (Sitch et al., 2008). Our results follow the consistent pattern simulated by these other models, but our unique experimental design allows us to isolate the impact of dynamic land cover implementation (vs. prescribed land cover implementation) and compare it against the effects of other global change drivers.

## 5. Conclusions

Our results illustrate the potential effects of range shifts alongside variation in plant growth that is driven by climate change. While all terrestrial biosphere models account for the influence of global change drivers such as $CO_2$ fertilisation, increasing temperatures, and variable precipitation regimes on plant growth, terrestrial biosphere models with prescribed land cover cannot capture range shifts that are driven by climate change. However, few terrestrial biosphere models incorporate dynamic land cover. Importantly, the shifting ranges of biomes have critical feedbacks to climate change as well as critical consequences for biodiversity and human well-being (Pecl et al., 2017) alongside their consequences to the global C cycle.

## Appendix A: Detailed description of dynamic land cover in CLASSIC.

When dynamic land cover is implemented, the fractional coverage of a natural PFT is the result of colonization and mortality and is described in detail in Melton & Arora (2016). The colonization rate of PFT $n$ ($c_n$; day$^{-1}$) is determined by its net
primary productivity (NPP):

$$c_n = \min\left[\max\left[\lambda_{1,n}, \lambda_{2,n}\right], 0.1\right] NPP_n \frac{1}{s_{sap,n} \max\left[0.25, \min\left[C_{veg,n}, 5.0\right]\right]} \tag{A1}$$

$NPP_n$ is the NPP of PFT $n$ and $C_{veg,n}$ is the vegetation C biomass of PFT $n$. $s_{sap,n}$ is a factor for converting vegetation C biomass to seedling C biomass of PFT $n$ (unitless) (Table B3). $\lambda_{1,n}$ and $\lambda_{2,n}$ determine the fraction of NPP that is used for spatial expansion within a given grid cell of PFT $n$ (unitless):

$$
\quad \lambda_{1,n} = \begin{cases} 0, & LAI_n \leq LAI_{min,n} \\ 0.1\frac{LAI_n - LAI_{min,n}}{LAI_{max,n} - LAI_{min,n}}, & LAI_{min,n} < LAI_n < LAI_{max,n} \\ 0.1, & LAI_{max,n} \leq LAI_n \end{cases} \tag{A2}
$$

$$
\lambda_{2,n} = \begin{cases} 0, & LAI_n \leq 0.25 LAI_{min,n} \\ \cosh\left(0.115\left(LAI_n - 0.25 LAI_{min,n}\right)\right) - 1, & 0.25 LAI_{min,n} < LAI_n \end{cases} \tag{A3}
$$

$LAI_{min,n}$ and $LAI_{max,n}$ are minimum and maximum leaf area index thresholds, respectively, of PFT $n$ (m$^2$ m$^{-2}$) (Table B3). The mortality rate of PFT $n$ ($m_n$; day$^{-1}$) is the sum of intrinsic or age-related mortality ($m_{intr,n}$; day$^{-1}$), mortality due to reduced growth ($m_{ge,n}$; day$^{-1}$), mortality due to fire ($m_{dist,n}$; day$^{-1}$), and mortality when a PFT exists outside its bioclimatic
limits ($m_{bioclim,n}$; day$^{-1}$):

$$m_n = m_{intr,n} + m_{ge,n} + m_{dist,n} + m_{bioclim,n} \tag{A4}$$

$$m_{intr,n} = 1 - \exp\left(\frac{-4.605}{A_{max,n}}\right) \tag{A5}$$

$$m_{ge,n} = \frac{m_{ge,max,n}}{1 + \left(300 \, m^2/kg \, C\right)\left(\max[0, ge_n]\right)} \tag{A6}$$

$$m_{dist,n} = \varsigma_{r,n} \frac{f_{burned,n}}{f_n} \tag{A7}$$

$A_{max,n}$ is the maximum age of PFT $n$ (yr) (Table B3). $m_{ge,max,n}$ is the maximum growth-related mortality rate occurring when no growth occurs of PFT $n$ (day$^{-1}$) (Table B3). $ge_n$ is growth efficiency over the previous year of PFT $n$ (kg C m$^{-2}$):

$$ge_n = \frac{\Delta C_{s,n} + \Delta C_{r,n}}{L_{max,n}} \tag{A8}$$

$\Delta C_{s,n}$ is the stem C increment over the previous year of PFT $n$ (kg C m$^{-2}$), $\Delta C_{r,n}$ is the root C increment over the previous year of PFT $n$ (kg C m$^{-2}$), and $L_{max,n}$ is the maximum LAI over the previous year of PFT $n$ (m$^2$ m$^{-2}$). $\varsigma_{r,n}$ is the susceptibility
to stand replacing fire of PFT $n$ (fraction) (Table B3). $f_{burned,n}$ is the fractional area burned within a day of PFT $n$ and depends on wind speed and soil moisture as well as the probability of fire occurrence (which depends on the availability of vegetation biomass as a fuel source, the combustibility of the fuel source based on soil moisture, and the presence of an

ignition source based on lightning and population density) and suppression (which depends on population density). This is described in detail in Arora & Melton (2018).

Six bioclimatic indices are used to determine the spatial range of PFTs, representing the physiological limits to their survival: air temperature of the coldest month, air temperature of the warmest month, aridity index (ratio of potential evaporation to precipitation), growing degree days (cumulative number of days with air temperature above 5°C), dry season length (number of consecutive months with precipitation less than potential evaporation), and precipitation surplus (difference between precipitation and potential evaporation). Bioclimatic indices for each grid cell are updated annually on a

25-year timescale using exponential smoothing:

$$X(t + 1) = X(t)e^{-1/25} + x(t)\left(1 - e^{-1/25}\right) \tag{A9}$$

$x(t)$ represents a bioclimatic index at year $t$ and $X(t)$ represents the smoothed bioclimatic index at year $t$. This accounts for time lags in the response of vegetation to climate change drivers (Wu et al., 2015). At the beginning of each time step, each grid cell is assigned with a small fractional coverage of each PFT (0.001). Whether, the PFT persists is determined by

$m_{bioclim,n}$: $m_{bioclim,n} = 0.25$ when any bioclimatic index is outside its corresponding bioclimatic limit for a given PFT and $m_{bioclim,n} = 0$ when all bioclimatic indices are inside their corresponding bioclimatic limits for a given PFT. Bioclimatic limits for each PFT are given in Table B1. These were derived by first calculating the bioclimatic indices for each grid cell using the CRUJRA climate dataset averaged over 1900 to 1920 (Harris et al., 2020). Then, the 1% and 99% quantiles of these bioclimatic indices over all grid cells for each land cover type in the Collection 5 MODIS Global Land Cover Type

International Geosphere-Biosphere Programme (IGBP) product (Friedl et al., 2010) were calculated and associated with a CLASSIC PFT. Note that this assumes that the current biome ranges are in equilibrium with the 1900-1920 climate.

Each natural PFT is given a dominance rank ($i$) where PFTs with higher dominance ranks invade PFTs with lower dominance ranks and bare ground. Natural PFTs cannot invade crop area. Tree PFTs are given higher dominance ranks than grass PFTs. Within tree PFTs and within grass PFTs, a PFT with a higher $c_n$ is dominant over all PFTs with lower $c_n$. Each

PFT is thus ranked from $1, 2, \dots, i-1, i, i+1, \dots N$ where the PFT with dominance rank 1 is the most dominant. Overall, the change in fractional coverage of a PFT with dominance rank $i$ is due to (1) its colonization into the areas of PFTs with lower dominance ranks ($i+1, \dots, N$) and bare ground ($N+1$), (2) colonization by PFTs with higher dominance ranks ($i, \dots, i-1$) into its area, and (3) its mortality:

$$\frac{df_i}{dt} = (c_i f_{i+1} + \dots + c_i f_N + c_i f_{N+1}) - (c_1 f_i + \dots + c_{i-1} f_i) - m_i f_i \tag{A10}$$

The change in fractional coverage of bare ground is due to mortality of the PFTs and colonization of bare ground by the PFTs:

$\frac{df_{N+1}}{dt} = \sum_{n=1}^{N} m_n f_n - \sum_{n=1}^{N} c_n f_{N+1}$                                                      (A11)

**Appendix B: Supplemental figures and tables.**

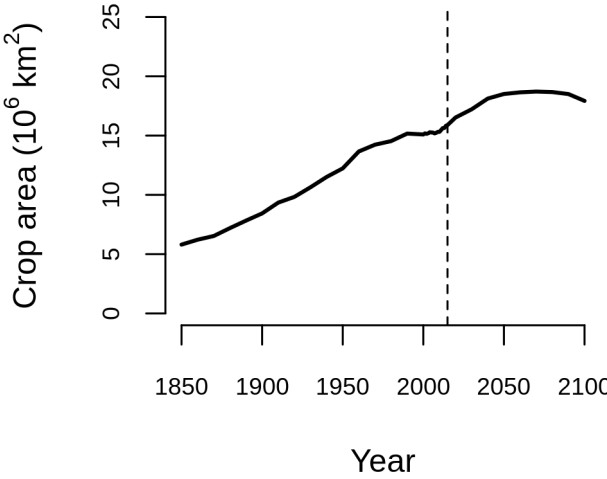

**Figure B1: Crop area over the historical period (1850 – 2015) and SSP5-8.5 (2015 – 2100) from the LUH2-GCB2022 (Land-Use Harmonization 2) dataset (Chini et al., 2021; Hurtt et al., 2020; Klein Goldewijk et al., 2017).**

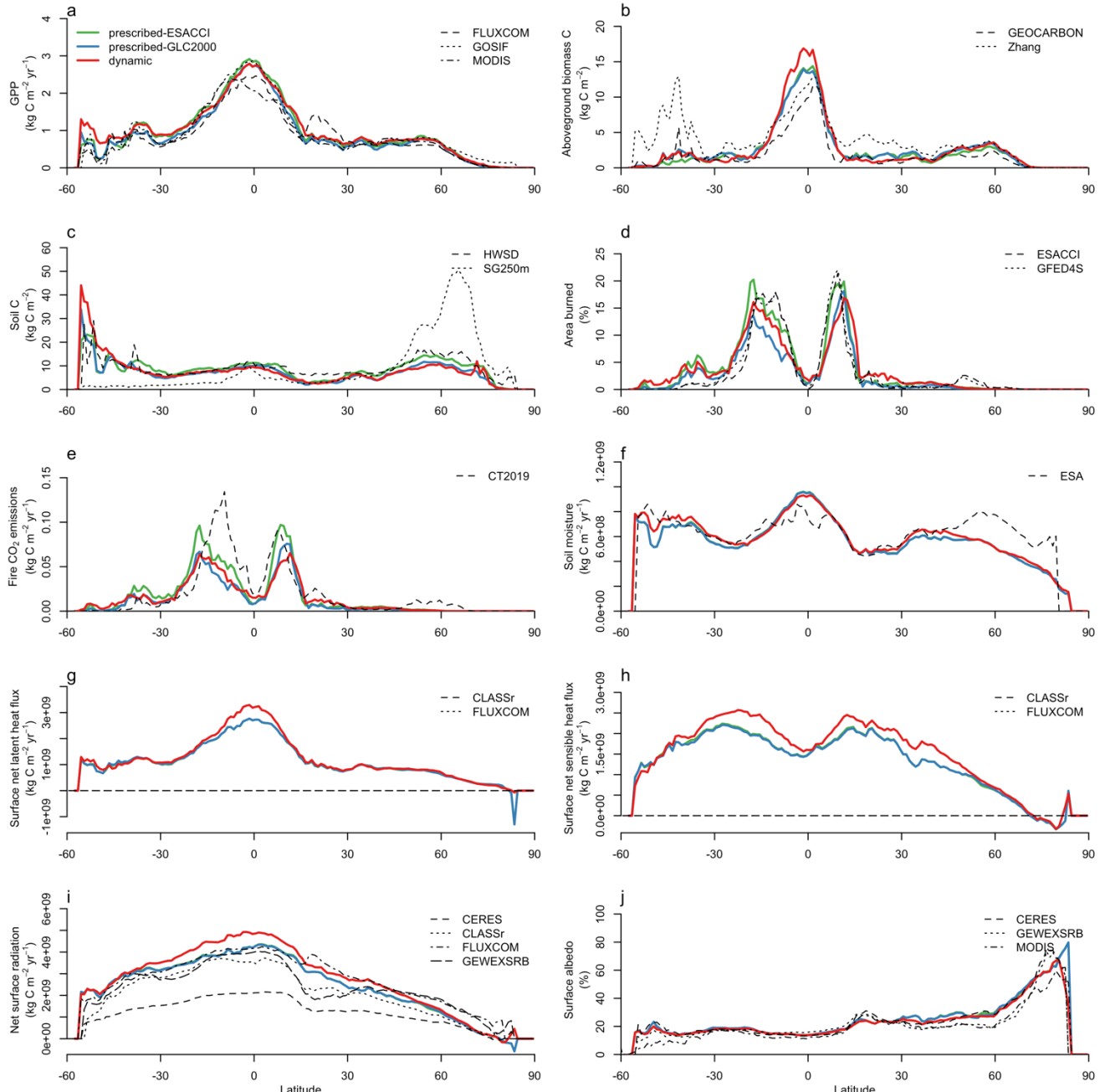

**Figure B2: Latitudinal distributions of a. gross primary productivity (GPP), b. aboveground biomass C, c. soil C, d. area burned, e. fire CO$_2$ emissions, f. soil moisture, g. surface net latent heat flux, h. surface net sensible heat flux, i. net surface radiation, and j. surface albedo simulated by CLASSIC with prescribed land cover (with an ESACCI-derived land cover forcing and a GLC2000-derived land cover forcing) and CLASSIC with dynamic land cover in comparison to present-day observations. CLASSIC simulations were averaged over 2000-2020. Observation-based datasets are described in the Methods. Figure B2 shows the scores of model performance in reproducing observation-based datasets for each variable.**



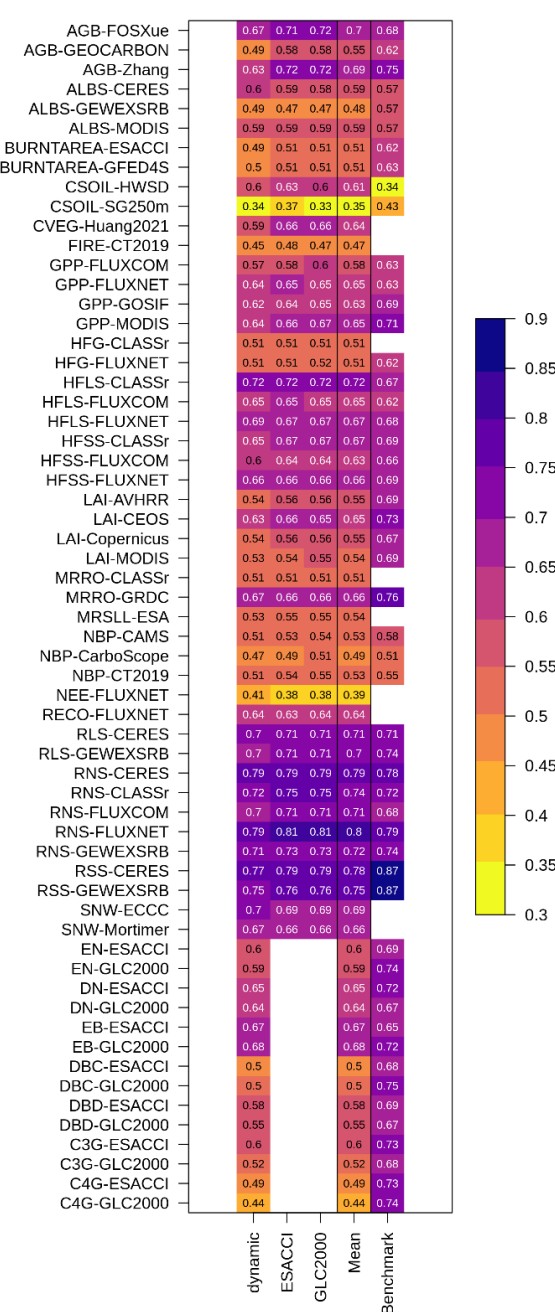

**Figure B3: Scores of model performance in reproducing observation-based datasets of principal C, water, and energy variables as well as land cover fractions of each natural plant functional type (PFT) for simulations of CLASSIC with prescribed land cover (with an ESACCI-derived land cover forcing and a GLC2000-derived land cover forcing) and dynamic land cover. Abbreviations for C, water, and energy variables and observation-based datasets are described in the Methods. Abbreviations for PFTs are evergreen needleleaf (EN), deciduous needleleaf (DN), evergreen broadleaf (EB), deciduous broadleaf cold (DBC), deciduous broadleaf dry (DBD), $C_3$ grass (C3G), and $C_4$ grass (C4G).**


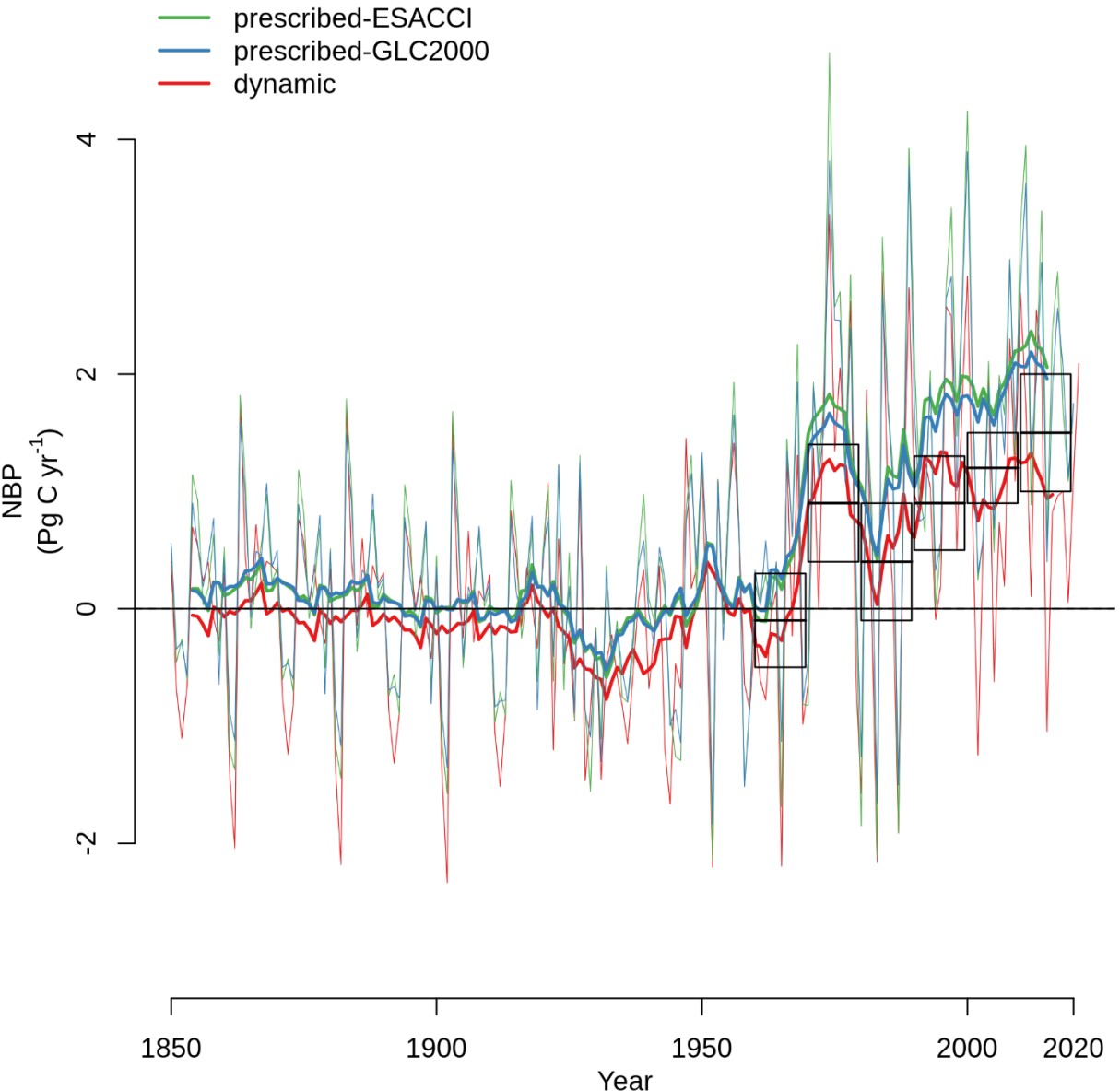

**Figure B4: Net biome productivity (NBP) simulated by CLASSIC with prescribed land cover (with an ESACCI-derived land cover forcing and a GLC2000-derived land cover forcing) and CLASSIC with dynamic land cover. Boxes indicate estimates from the suite of models in the Global Carbon Project where dark horizontal lines indicate the mean and the extent indicates one standard deviation (Friedlingstein et al., 2022). Thin lines indicate the annual value and thick lines indicate the moving average over 10 years.**

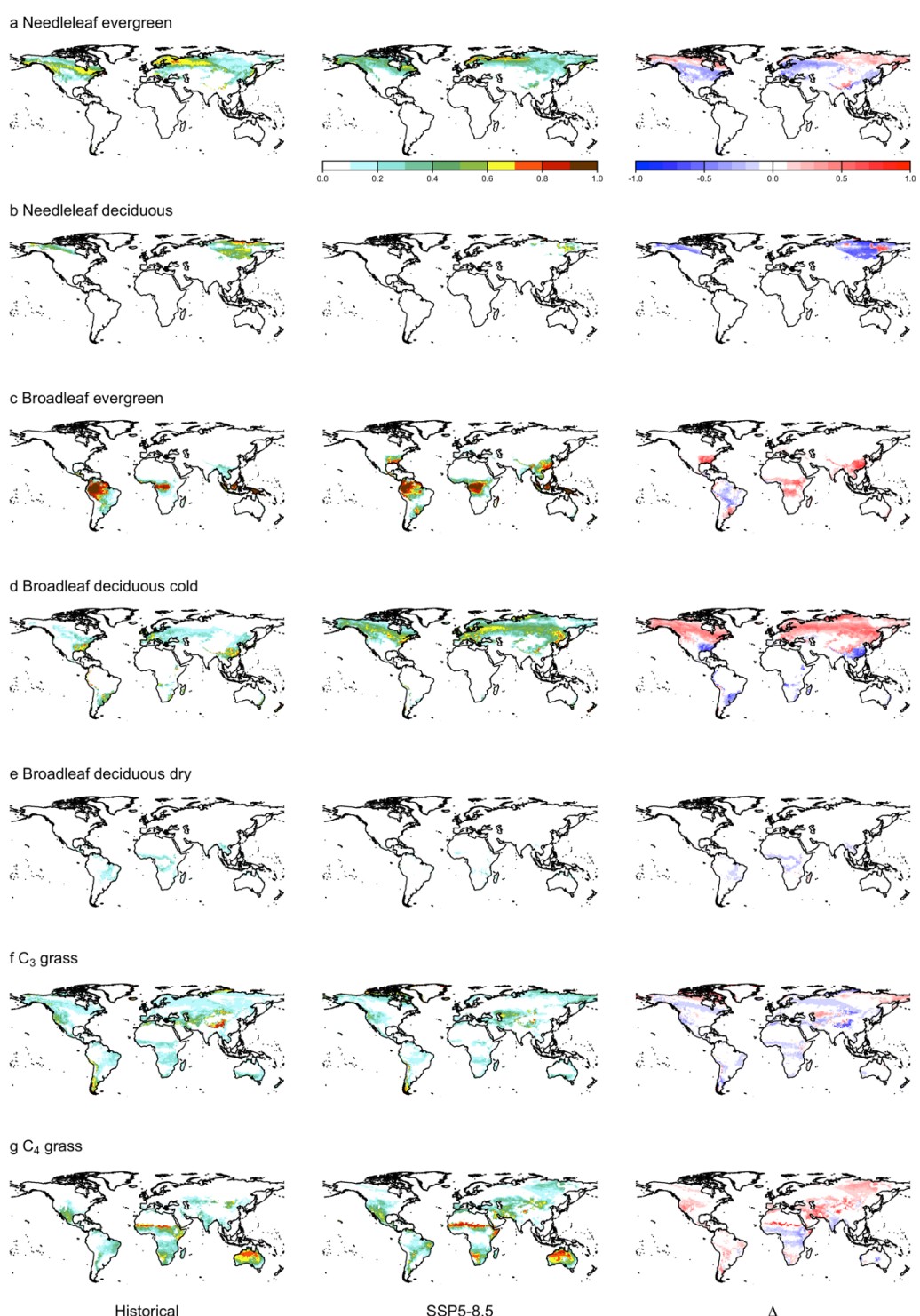

**Figure B5: Fraction per grid cell in the historical period (averaged over 1995-2015), in SSP5-8.5 (averaged over 2080-2100), and change between the historical period and SSP5-8.5 in for each plant functional type (PFT) simulated by CLASSIC with dynamic**

land cover. **a. Evergreen needleleaf, b. deciduous needleleaf, c. evergreen broadleaf, d. deciduous broadleaf cold, e. deciduous broadleaf dry, f. C$_3$ grass, and g. C$_4$ grass. Differences reflect the change between the average over 1995-2015 and 2080-2100.**

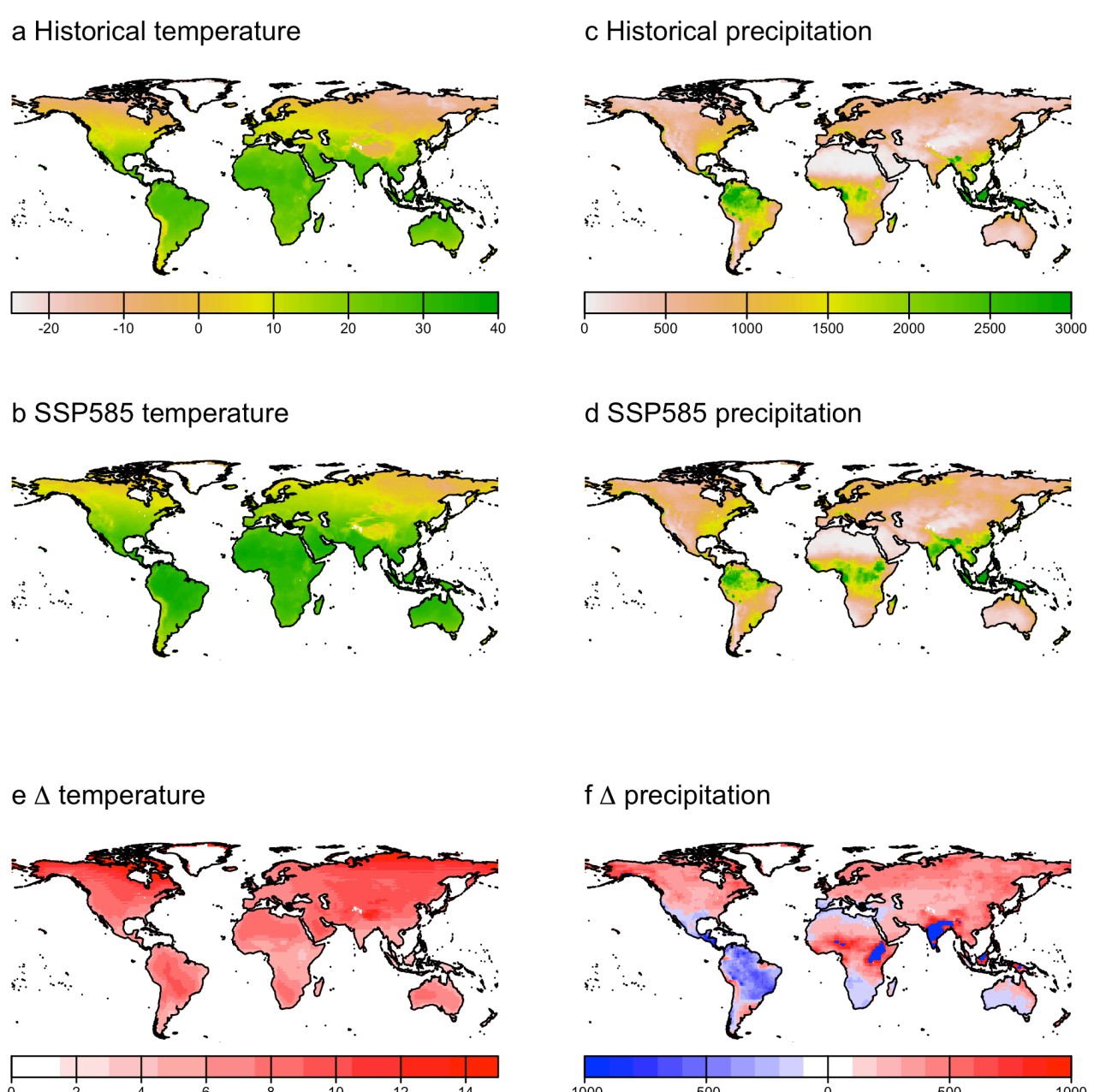


**Figure B6: Temperature and precipitation in ab. the historical period (averaged over 1995-2015), cd. SSP5-8.5 (averaged over 2080-2100), and ef. change between the historical period and SSP5-8.5.**

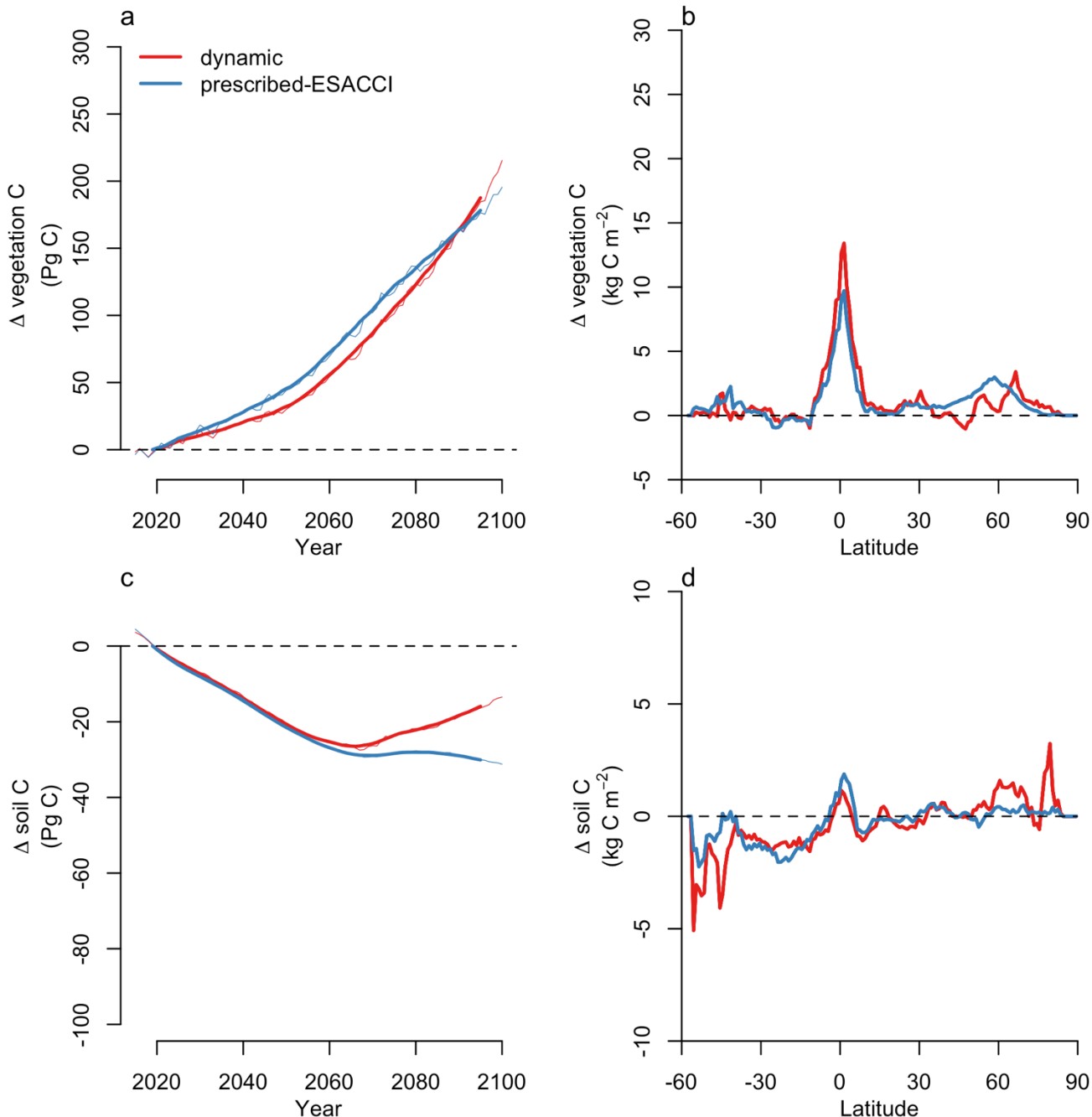

**Figure B7: Change in ab. vegetation C and cd. soil C simulated by CLASSIC with prescribed land cover (with an ESACCI-derived land cover forcing) and CLASSIC with dynamic land cover for SSP5-8.5 (2015 – 2100). For the latitudinal distributions, CLASSIC simulations were averaged over 2080-2100. Thin lines indicate the annual value and thick lines indicate the moving average over 10 years.**

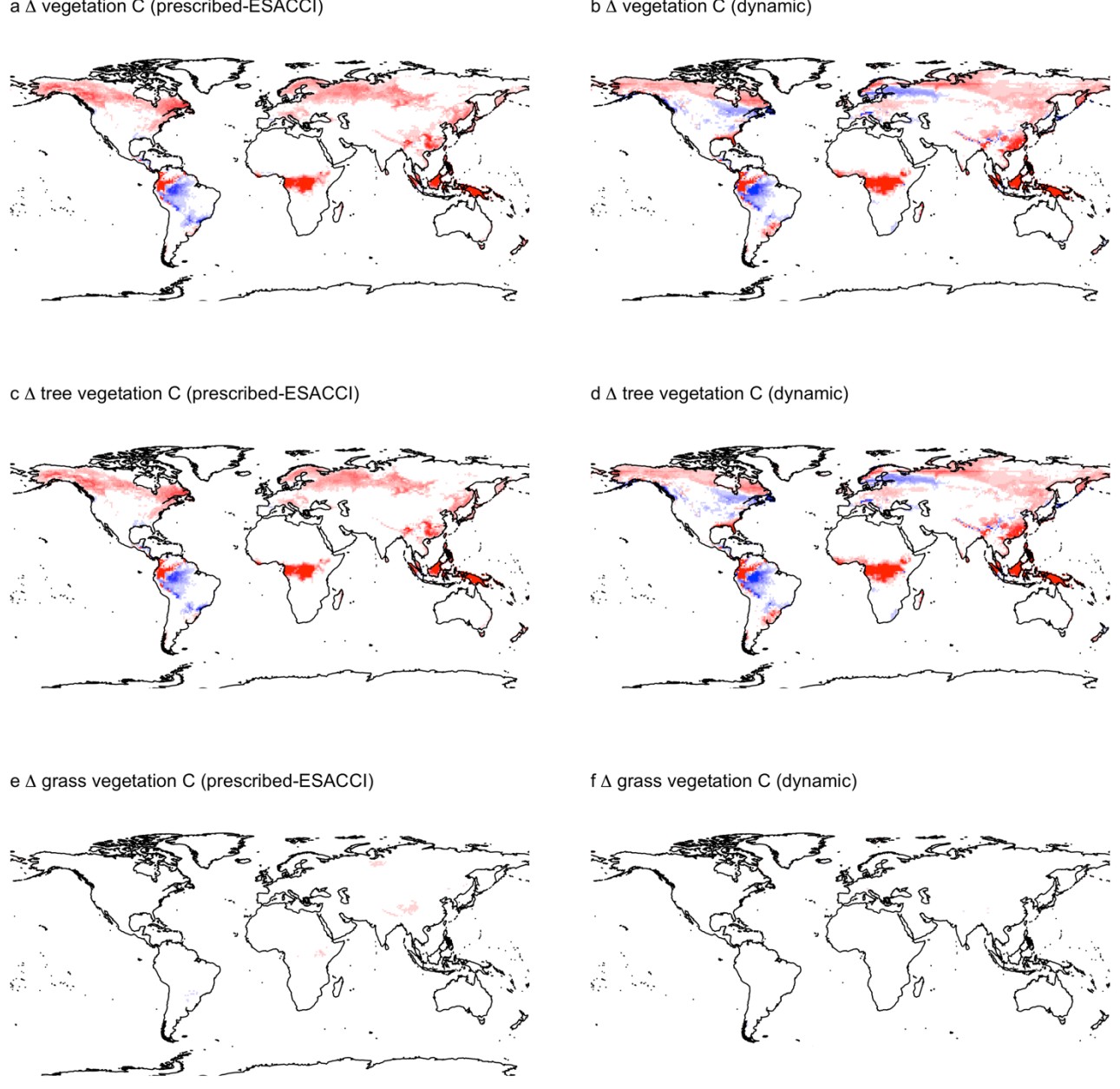

**Figure B8: Change in vegetation C simulated by CLASSIC with prescribed land cover (with an ESACCI-derived land cover forcing) and CLASSIC with dynamic land cover for SSP5-8.5 (2015 – 2100). ab. Total vegetation C, cd. tree vegetation C, and ef. grass vegetation C. Differences reflect the change between the average over 1995-2015 and 2080-2100.**

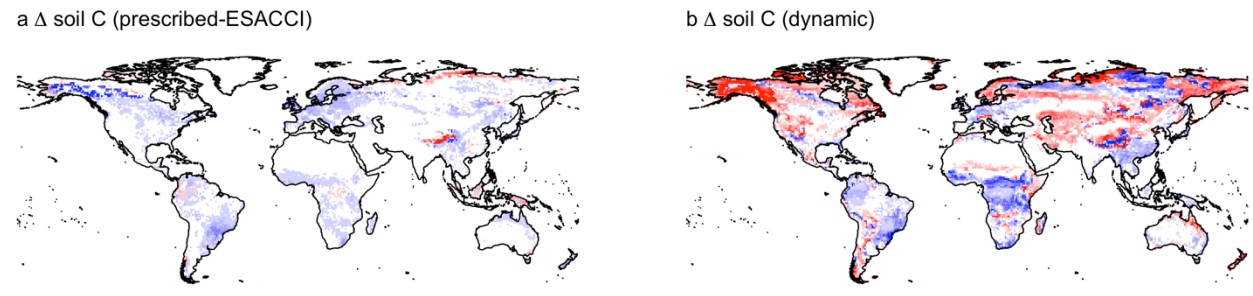

**Figure B9: Change in soil C simulated by a. CLASSIC with prescribed land cover (with an ESACCI-derived land cover forcing) and b. CLASSIC with dynamic land cover for SSP5-8.5 (2015 – 2100). Differences reflect the change between the average over 1995-2015 and 2080-2100.**


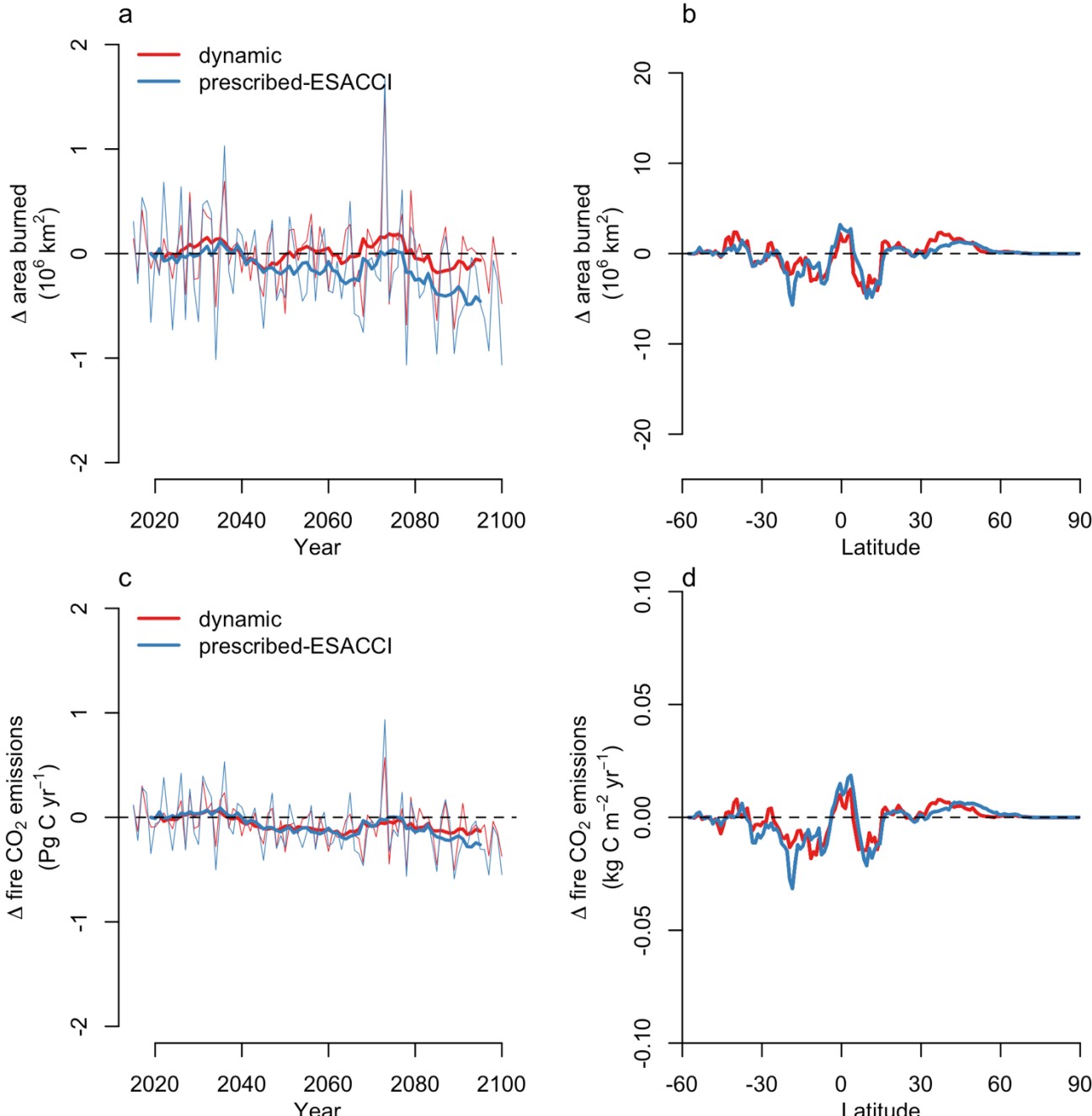

**Figure B10: Change in ab. area burned and cd. fire CO₂ emissions by CLASSIC with prescribed land cover (with an ESACCI-derived land cover forcing) and CLASSIC with dynamic land cover for SSP5-8.5 (2015 – 2100). For the latitudinal distributions, CLASSIC simulations were averaged over 2080-2100. Thin lines indicate the annual value and thick lines indicate the moving average over 10 years.**

a Δ area burned (prescribed-ESACCI)

b Δ area burned (dynamic)

c Δ tree area burned (prescribed-ESACCI)

d Δ tree area burned (dynamic)

e Δ grass area burned (prescribed-ESACCI)

f Δ grass area burned (dynamic)

**Figure B11: Change in area burned simulated by CLASSIC with prescribed land cover (with an ESACCI-derived land cover forcing) and CLASSIC with dynamic land cover for SSP5-8.5 (2015 – 2100). ab. Total area burned, cd. tree area burned, and ef. grass area burned. Differences reflect the change between the average over 1995-2015 and 2080-2100.**

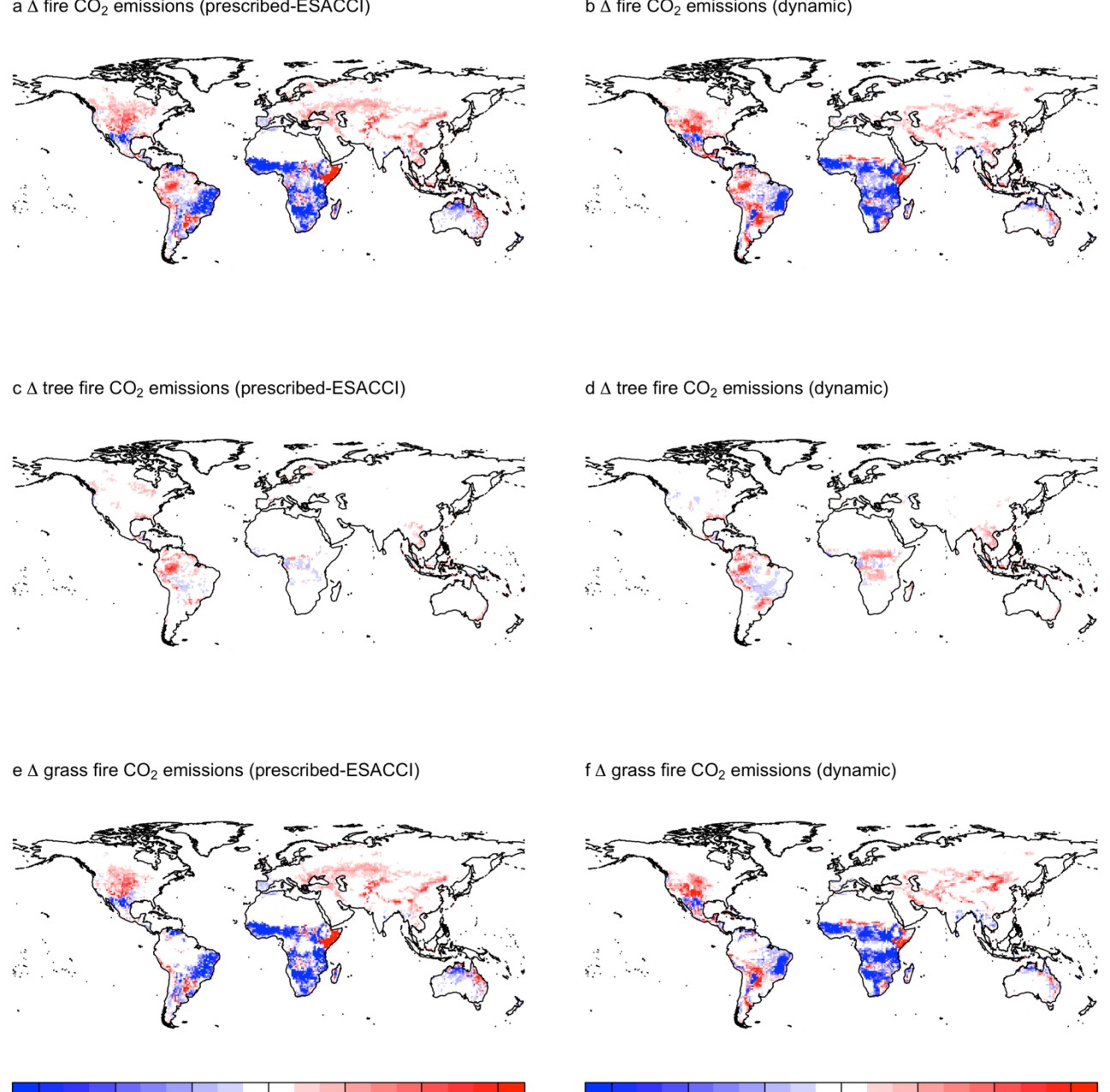


**Figure B12: Change in fire CO₂ emissions simulated by CLASSIC with prescribed land cover (with an ESACCI-derived land cover forcing) and CLASSIC with dynamic land cover for SSP5-8.5 (2015 – 2100). ab. Total fire CO₂ emissions, cd. tree fire CO₂ emissions and ef. grass fire CO₂ emissions. Differences reflect the change between the average over 1995-2015 and 2080-2100.**

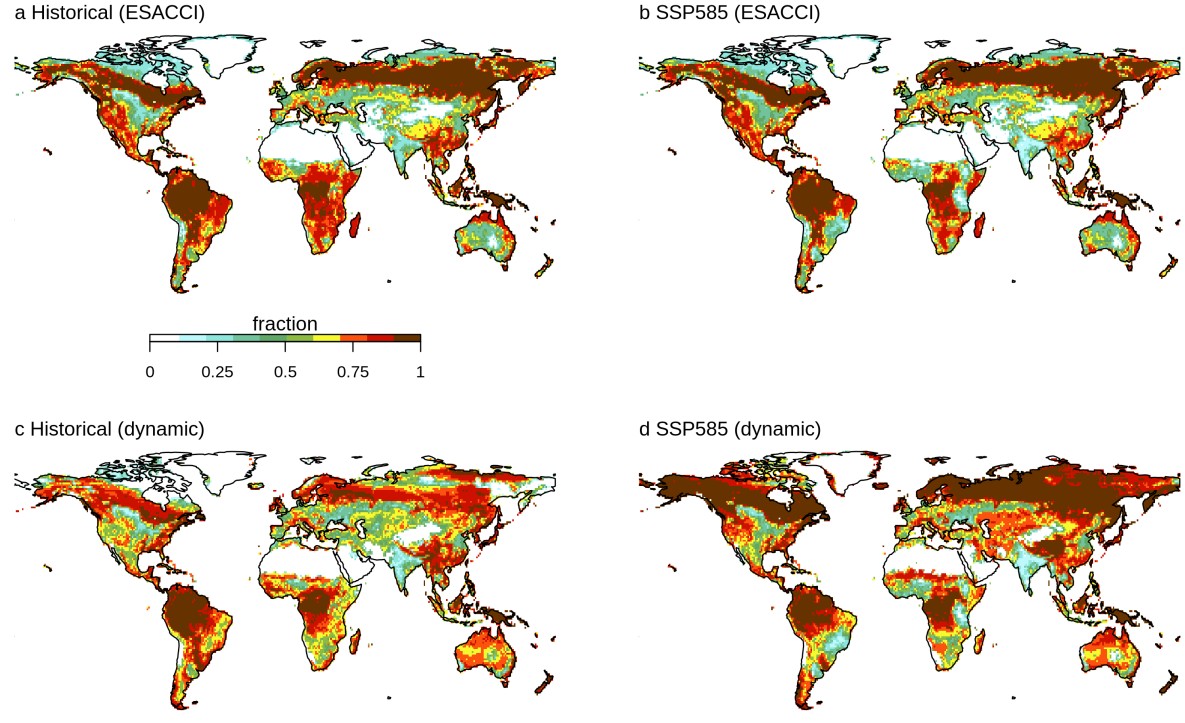

Figure B13: Natural vegetation fraction per grid cell ab. in the ESACCI-derived land cover product and cd. simulated by CLASSIC with dynamic land cover over ac. the historical period (averaged over 1995-2015) and bd. for SSP5-8.5 (averaged over 2080-2100).

**Table B1: Parameters associated with bioclimatic indices.**

| Parameter | EN | DN | EB | DBC | DBD | C3G | C4G |
|---|---|---|---|---|---|---|---|
| Maximum air temperature of the coldest month | -2 | -27 | | 17 | | | |
| Minimum air temperature of the coldest month | | | 1 | -33 | 15 | | |
| Maximum air temperature of the warmest month | 28 | 19 | | 30 | | | |
| Minimum air temperature of the warmest month | | | | | | | 20 |
| Maximum aridity index | 8 | 8 | 8 | 8 | 8 | 8 | 8 |
| Minimum aridity index | | | | | 0.5 | | |
| Minimum growing degree days | 370 | 340 | 1320 | 430 | 4920 | | |
| Maximum dry season length | | | 8 | | | | |
| Minimum dry season length | | | | | 5 | | |

| Precipitation surplus | 150 | 150 | 150 | 150 | 150 |
|---|---|---|---|---|---|


**Table B2: Observation-based datasets used in AMBER. Described in detail in (Seiler et al., 2022). Datasets are globally grided or in situ (specified).**

| Variable | Variable abbreviation | Observation-based dataset |
|---|---|---|
| Aboveground biomass carbon | AGB | FOSXue (in situ) (Schepaschenko et al., 2019; Xue et al., 2017) GEOCARBON (Avitabile et al., 2016; Santoro et al., 2015) Zhang (Zhang and Liang, 2020) |
| Albedo | ALBS | CERES (Kato et al., 2013) GEWEXSRB (Stackhouse et al., 2011) MODIS (Strahler and Muller, 1999) |
| Area burned | BURNTAREA | ESACCI (Chuvieco et al., 2018) GFED4S (Giglio et al., 2010) |
| Soil carbon | CSOIL | HWSD (Todd-Brown et al., 2013; Wieder, 2014) SG250m (Hengl et al., 2017) |
| Vegetation carbon | CVEG | Huang2021 (Huang et al., 2021) |
| Fire emissions | FIRE | CT2019 (Jacobson et al., 2020) |
| Gross primary productivity | GPP | FLUXCOM (Jung et al., 2020) GOSIF (Li and Xiao, 2019) MODIS (Zhang et al., 2017) FLUXNET (in situ) (Pastorello et al., 2020) |
| Soil heat flux | HFG | CLASSr (Hobeichi et al., 2020) FLUXNET (in situ) (Pastorello et al., 2020) |
| Latent heat flux | HFLS | CLASSr (Hobeichi et al., 2020) FLUXCOM (Jung et al., 2019) FLUXNET (in situ) (Pastorello et al., 2020) |
| Sensible heat flux | HFSS | CLASSr (Hobeichi et al., 2020) FLUXCOM (Jung et al., 2019) FLUXNET (in situ) (Pastorello et al., 2020) |
| Leaf area index | LAI | AVHRR (Claverie et al., 2016) CEOS (in situ) (Garrigues et al., 2008) Copernicus (Verger et al., 2014) MODIS (Myneni et al., 2002) |
| Streamflow | MRRO | CLASSr (Hobeichi et al., 2020) GRDC (in situ) (Dai and Trenberth, 2002) |
| Soil moisture | MRSLL | ESA (Liu et al., 2011) |
| Net biome productivity | NBP | CAMS (Agustí-Panareda et al., 2019) CarboScope (Rödenbeck et al., 2018) CT2019 (Jacobson et al., 2020) |
| Net ecosystem exchange | NEE | FLUXNET (in situ) (Pastorello et al., 2020) |
| Ecosystem respiration | RECO | FLUXNET (in situ) (Pastorello et al., 2020) |
| Net surface long wave radiation | RLS | CERES (Kato et al., 2013) GEWEXSRB (Stackhouse et al., 2011) |
| Net surface radiation | RNS | CERES (Kato et al., 2013) CLASSr (Hobeichi et al., 2020) FLUXCOM (Jung et al., 2019) FLUXNET (Pastorello et al., 2020) GEWEXSRB (Stackhouse et al., 2011) |

| | | | | | | |
|---|---|---|---|---|---|---|
| Net surface shortwave radiation | RSS | CERES (Kato et al., 2013) | | | | |
| | | GEWEXSRB (Stackhouse et al., 2011) | | | | |
| Snow water equivalent | SNW | Mortimer (Mortimer et al., 2020) | | | | |

**Table B3: Parameters associated with the calculation of colonization and mortality rates for each PFT.**

| Parameter | EN | DN | EB | DBC | DBD | C3G | C4G |
|---|---|---|---|---|---|---|---|
| $s_{sap,n}$ | 0.30 | 0.10 | 0.10 | 0.14 | 0.30 | 0.10 | 0.10 |
| $LAI_{min,n}$ | 1.0 | 0.25 | 1.5 | 1.5 | 1.5 | 0.5 | 0.5 |
| $LAI_{max,n}$ | 4.0 | 2.0 | 6.0 | 6.0 | 6.0 | 3.0 | 3.0 |
| $A_{max,n}$ | 250 | 400 | 600 | 250 | 500 | 0 | 0 |
| $m_{ge,max,n}$ | 0.005 | 0.005 | 0.005 | 0.005 | 0.005 | 0.1 | 0.1 |
| $\varsigma_{r,n}$ | 0.20 | 0.20 | 0.50 | 0.20 | 0.15 | 0.25 | 0.25 |

## Code availability

The source code for CLASSIC is available on the CLASSIC community Zenodo page (https://zenodo.org/communities/classic?q=&l=list&p=1&s=10&sort=newest). AMBER is available at https://gitlab.com/cseiler/AMBER.

## Author contribution

S.K.G. designed and conducted the study and prepared the initial manuscript. V.K.A. provided input on the study design and the initial manuscript. C.S. and L.W. provided input on the subsequent revisions.

## Competing interests

The authors declare that they have no conflict of interest.

## Acknowledgements

The authors would like to thank the input from the CLASSIC team (Paul Bartlett, Mike Brady, Ed Chan, Salvatore Curasi, Joe Melton, and Gesa Meyer).

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
