# Peer review of "The impacts of modelling prescribed vs. dynamic land cover in a high CO2 future scenario – greening of the Arctic and Amazonian dieback"

_EGUsphere, 2023_

## Author Response (AR1)

*AUTHOR COMMENTS: We thank both referees for their insightful comments. We appreciate that the reviewers were generally favourable towards the manuscript. Their comments were extremely helpful in improving our study. Below we provide point-by-point responses to these comments.*
* * *
Referee #1:

The study investigates the impact of dynamically modeling land cover versus prescribing it in future high CO2 scenarios, focusing on Arctic greening and Amazonian dieback. It emphasizes the significance of simulating dynamic land cover to project future terrestrial carbon sinks accurately. Comparing simulations with prescribed and dynamic land cover, the study reveals substantial differences in predicting terrestrial carbon sinks, with dynamic simulations indicating a larger sink due to biome shifts such as tree expansion into the Arctic and the Amazon transitioning from forest to grassland. This highlights the crucial role of climate change-driven biome shifts in forecasting future carbon dynamics.

This issue reflects the reason why Dynamic Global Vegetation Models (DGVMs) were developed. There has been considerable research in this area, and the novelty claimed by the authors may not be as significant as suggested. However, there is value in accumulating such studies, and I do not oppose the publication of this manuscript. Below are some technical comments.

*AUTHOR COMMENTS: We agree that this is a well-studied research area, and we hope that we have effectively described the background and previous research conducted. We think that we have applied a novel approach (in comparing the same model with prescribed vs. dynamic land cover thereby isolating the influence of dynamic land cover) and by introducing a new framework to evaluate model performance in simulating land cover itself to study this topic. We have highlighted this in Lines 15-19 of the revised manuscript. As the reviewer points out, our results align with the results of previous studies, building on and contributing to the accumulation of such studies. Additionally, as we point out in Lines 425-426, only 3 out of 11 contributing models to C4MIP within the 6th phase of CMIP implement dynamic land cover, highlighting the relevance of this research.*

Minor issues:

(1) Line157 "Mortality when a PFT exists outside its bioclimatic limits"
Can PFTs exist outside of their bioclimatic limits?

*AUTHOR COMMENTS: Thank you for pointing out that this was unclear. We have clarified mortality when a PFT exists outside its bioclimatic limits in Lines 163-175. Briefly, this mortality ensures that PFTs do not venture outside of their bioclimatic envelopes. Each grid cell is "seeded" with a small fractional coverage of each natural PFT (0.001). Whether a given PFT persists is determined by its bioclimatic limits. If the PFT is within its bioclimatic limits, there is no mortality associated with bioclimatic limits (m = 0) and the PFT can persist and expand. If the PFT is outside of its bioclimatic limits, the PFT is "killed off" by mortality associated with bioclimatic limits (m = 0.25).*

(2) Line165 "$m_{bioclim,n}$"
There is no explanation for how to use this index.

AUTHOR COMMENTS: *$m_{bioclim,n}$ is the mortality when a PFT exists outside its bioclimatic limits for PFT n. We have explained this in Line 163 of the revised manuscript.*

(3) Line175 "$c_n$"
How is this variable calculated?

AUTHOR COMMENTS: *The calculation of this variable is described in detail in Appendix A and briefly described in the main text. We have added a reference to Appendix A in Lines 154 and 156.*

(4) Lines230-231 "We conducted different pre-industrial spin ups for each simulation"
How were they different?

AUTHOR COMMENTS: *We have clarified this in Lines 229-234. We conducted a different pre-industrial spin-up for simulations with prescribed landcover and for simulations with dynamic landcover. Within the simulations for prescribed landcover, we conducted different pre-industrial spin-ups for simulations with GLC2000 land cover and ESACCI land cover. These are described in Table 1.*

(5) Figure 1a
This figure compares the temporal changes in the ratio of natural vegetation areas. Is it correct to understand that the ratio of natural vegetation area mentioned here excludes deserts, ice sheets, and croplands from the terrestrial area? It is necessary to define this term in the figure caption or the main text.

AUTHOR COMMENTS: *Thank you for pointing out that this was unclear. All simulations exclude Antarctica and Greenland. We have clarified this in Lines 147-148 and 233-234. We have also clarified that crop area and bare ground are excluded in the figure caption (as well as in the caption for Figure 4).*
* * *
Referee #2:

General comments:

Kou Giesbrecht et al. utilize a dynamic land cover model to understand how changes in land cover can change the carbon sink in high CO2 future scenarios. The authors argue that implementing a dynamic land cover model changes the carbon fluxes where at the end of the century the net biome productivity is twice as large with the dynamic land cover than without it at the end of the century. While other studies have included dynamic land models, this research is still highly relevant and important to emphasize as many CMIP6 models still do not include dynamic vegetation and/or fires which can have impacts on the future land carbon sink as the

authors have described in this paper. While this reviewer believes that this study is worth publishing, there are a few points this reviewer thinks would be important to address prior to the publication. These points of discussion include more discussions on the role of fire in the changing land cover, carbon dynamics as well as the changes in future carbon storage. There are also questions regarding the statistical significance of changes of vegetation/PFT which this reviewer thinks may be important to note in their study. This reviewer believes there should also be some clarifications in the methods regarding the year of data used from various products such as ESA CCI and the MODIS.

*AUTHOR COMMENTS: Thank you for your positive review. We have addressed these individual points below. We have also emphasized the relevance of our work for model intercomparison projects in Lines 16-19.*

Specific comments:

1) When creating the bioclimatic index, the authors note on L171-2 that this assumes the current biome ranges are in equilibrium with the 1900-1920 climate. There have been many studies recently, particularly focused on the high latitudes in Alaska and Canada, showing that the land cover has changed substantially since 1985, as increasing burned areas from wildfire continue to change the land cover (Macander et al. 2022, Wang et al., 2019). Could the authors please elaborate/justify this assumption for the readers or have a small discussion around this issue. Macander, M. J., Nelson, P. R., Nawrocki, T. W., Frost, G. V., Orndahl, K. M., Palm, E. C., Wells, A. F., & Goetz, S. J. (2022). Time-series maps reveal widespread change in plant functional type cover across Arctic and boreal Alaska and Yukon. Environmental Research Letters, 17(5), 054042. https://doi.org/10.1088/1748-9326/ac6965 Wang, J. A., Sulla-Menashe, D., Woodcock, C. E., Sonnentag, O., Keeling, R. F., & Friedl, M. A. (2020). Extensive land cover change across Arctic–Boreal Northwestern North America from disturbance and climate forcing. Global Change Biology, 26(2), 807–822. https://doi.org/10.1111/gcb.14804

*AUTHOR COMMENTS: Thank you for drawing our attention to this issue. We have added a small discussion around this issue to Lines 180-184. Briefly, we assumed that current biome ranges are in equilibrium with the 1900-1920 climate when parameterising bioclimatic limits. We made this assumption because of the migration lag that occurs between climatic change and observed differences in established plant species ranges, especially in long-lived plant species such as trees (Corlett and Wescott 2013). While our approach captures slower shifts due to climatic change, it does not capture rapid shifts due to fire disturbance, as the reviewer points out (and as described in Macander et al. 2022 and Wang et al. 2020, which we have referenced in the revised manuscript).*

2) This reviewer noticed that C4 grasses increases in the Arctic boreal region in Figure B5. These areas usually only contain C3 grasses as seen by Still et al. 2003, which the authors cite, and a recent paper by Luo et al. 2024. One question this reviewer has is if the authors believe the increases in C4 grasses is a reasonable change in PFTs for the Arctic boreal region. This leads to another question of if the area change in PFTs is significant within the model. This reviewer

thinks it would be helpful to the readers for the authors to indicate areas of significant changes in their maps.

Luo, X., Zhou, H., Satriawan, T.W. et al. Mapping the global distribution of C4 vegetation using observations and optimality theory. Nat Commun 15, 1219 (2024). https://doi.org/10.1038/s41467-024-45606-3

*AUTHOR COMMENTS: Thank you for this keen observation and for pointing us to this recent paper. While high-latitude regions are currently dominated by C3 grasses, the increase in temperature that occurs in SSP5-8.5 could support the growth of C4 grasses previously not common at these latitudes. In alignment with both the crossover-temperature model and the optimality model which includes temperature in Luo et al. 2024, the temperature envelope of C4 grasses shifts northwards to encompass high-latitude regions in SSP5-8.5. We highlight this in Lines 359-360. We also have added absolute temperature (and precipitation) for the 1995-2015 average and for the 2081-2100 average for SSP5-8.5 to Figure B6, showing the similarities between the temperature envelope of C4 grasses in the historical period and in SSP5-8.5. We have also changed all maps to only show significant differences, which we have assessed using a Mann-Kendall trend test (P < 0.05).*

3) The authors mention that in the discussion that changes L407-419 the role that warming climate has on the expansion of vegetation. However as mentioned in an earlier comment, fires can accelerate this expansion allowing shrubs/deciduous trees to expand into these areas. For example, fires can replace the needleleaf evergreen conifers with more deciduous vegetation such as shrubs and/or deciduous forests (Baltzer et al. 2021, Liu et al. 2022, Lucash et al. 2023, Weiss et al. 2023). Can the authors expand on the effect that fires have within the model to explain the change in distribution of PFTs since fires are one of the primary mechanisms of change in the Arctic-boreal region?

Baltzer, J. L., Day, N. J., Walker, X. J., Greene, D., Mack, M. C., Alexander, H. D., Arseneault, D., Barnes, J., Bergeron, Y., Boucher, Y., Bourgeau-Chavez, L., Brown, C. D., Carrière, S., Howard, B. K., Gauthier, S., Parisien, M.-A., Reid, K. A., Rogers, B. M., Roland, C., … Johnstone, J. F. (2021). Increasing fire and the decline of fire adapted black spruce in the boreal forest. Proceedings of the National Academy of Sciences, 118(45), e2024872118. https://doi.org/10.1073/pnas.2024872118
Liu, Y., Riley, W.J., Keenan, T.F. et al. Dispersal and fire limit Arctic shrub expansion. Nat Commun 13, 3843 (2022). https://doi.org/10.1038/s41467-022-31597-6
Lucash, M. S., Marshall, A. M., Weiss, S. A., McNabb, J. W., Nicolsky, D. J., Flerchinger, G. N., Link, T. E., Vogel, J. G., Scheller, R. M., Abramoff, R. Z., & Romanovsky, V. E. (2023). Burning trees in frozen soil: Simulating fire, vegetation, soil, and hydrology in the boreal forests of Alaska. Ecological Modelling, 481, 110367. https://doi.org/10.1016/j.ecolmodel.2023.110367
Weiss, S. A., Marshall, A. M., Hayes, K. R., Nicolsky, D. J., Buma, B., & Lucash, M. S. (2023). Future transitions from a conifer to a deciduous-dominated landscape are accelerated by greater wildfire activity and climate change in interior Alaska. Landscape Ecology. https://doi.org/10.1007/s10980-023-01733-8

*AUTHOR COMMENTS: Thank you for drawing our attention to the role of fire, which we have elaborated on in the revised manuscript. First, we have explained that net biome productivity includes fire $CO_2$ emissions in Lines 405-406. Second, we have added a more*

*detailed explanation of the representation of fire in CLASSIC in Lines 158-162. In particular, we explain that area burned depends on a PFT-specific fire spread rate (where grasses have a higher fire spread rate than trees, and needleleaf trees have a higher spread rate than broadleaf trees). There are also additional differences between PFTs due to corresponding differences between vegetation biomass and soil moisture, which influence the probability of fire occurrence. We also clarify differences in the representation of fire between prescribed and dynamic land cover implementations in Lines 383-391. Briefly, when prescribed land cover is implemented, fire reduces vegetation biomass, whereas when dynamic land cover is implemented, fire both reduces vegetation biomass and creates bare ground that can then be colonized by a different plant functional type. Third, we added area burned and fire $CO_2$ emissions to Figure B2, which compares model output to observations over the historical period. Fourth, we have plotted area burned and fire $CO_2$ emissions in Figures B10, B11, and B12, which show global totals, latitudinal patterns, and maps. CLASSIC with dynamic land cover simulates slightly higher area burned and fire $CO_2$ emissions than CLASSIC with prescribed land cover due to higher natural vegetation area at the global scale (Figure B10), but spatial patterns are relatively similar between both versions of CLASSIC (Figure B11 and B12). In particular, neither version of CLASSIC simulates substantial changes in high latitude fires. It is a known issue that CLASSIC underestimates high latitude fires (Lines 387-388, Figure B2d, and see Arora and Melton, 2018), although this has been rectified in newer versions of CLASSIC (Curasi et al., 2024) and is currently in further development. A previous study showed that CLASSIC with dynamic land cover simulates the dominance of broadleaf trees in early succession followed by the dominance of evergreen trees later in succession from bare ground in boreal forests (Arora and Boer 2005, their Figure 5). Despite the low occurrence of high latitude fire, in our study CLASSIC simulates a transition from needleleaf to broadleaf trees in boreal forests. This is likely driven primarily by bioclimatic limits as well as fire. We have added broadleaf and needleleaf trees to Figure 6 to highlight this transition and added an explanation in Lines 369-371. We have also described the importance of how fires drive the transition from needleleaf to broadleaf trees in high latitude forests and cited the suggested studies on this topic in Lines 452-456.*

4) This reviewer thinks that  since the paper is describing the impacts of modeling prescribed vs. dynamic land cover, that the paper can benefit from authors discussing the changes in the future carbon storage and how changes in PFT change the overall carbon storage, expanding on the discussion of changes in productivity that is already included in the paper. The authors show that there is enhanced net biome productivity which would be anticipated with expansion of PFTs in high-latitudes. One question is if PFT conversions from one type to another, such as evergreen needleleaf to deciduous broadleaf, in high latitudes contribute to an enhancement of NBP within the model as well. How do the changes in NBP and PFT affect the distribution of carbon in the various pools from the beginning of the run if fires are combusting the vegetation/soils? There have been many studies showing an increase in primary productivity after fires from the flux tower level to remote sensing (Rocha and Shaver 2011, Coursolle et al. 2012, Kim et al. 2024), but a suppression of aboveground carbon sink (Wang et al. 2021). However, this can be counteracted with warming induced growth (Wang et al. 2023), as well as reduction in fire frequency with a change in PFT (Mack et al 2021). It would be interesting to expand on the discussion with the context of fire within the model.

Coursolle, C., Margolis, H. A., Giasson, M.-A., Bernier, P.-Y., Amiro, B. D., Arain, M. A., Barr, A. G., Black, T. A., Goulden, M. L., McCaughey, J. H., Chen, J. M., Dunn, A. L., Grant, R. F., & Lafleur, P. M. (2012). Influence of stand age on the magnitude and seasonality of carbon fluxes in Canadian forests. Agricultural and Forest Meteorology, 165, 136–148. https://doi.org/10.1016/j.agrformet.2012.06.011

Kim, J. E., Wang, J. A., Li, Y., Czimczik, C. I., & Randerson, J. T. (2024). Wildfire-induced increases in photosynthesis in boreal forest ecosystems of North America. Global Change Biology, 30(1), e17151. https://doi.org/10.1111/gcb.17151

Mack, M. C., Walker, X. J., Johnstone, J. F., Alexander, H. D., Melvin, A. M., Jean, M., & Miller, S. N. (2021). Carbon loss from boreal forest wildfires offset by increased dominance of deciduous trees. Science, 372(6539), 280–283. https://doi.org/10.1126/science.abf3903

Rocha, A. V., & Shaver, G. R. (2011). Burn severity influences postfire CO2 exchange in arctic tundra. Ecological Applications, 21(2), 477–489. https://doi.org/10.1890/10-0255.1

Wang, J., Taylor, A. R., & D'Orangeville, L. (2023). Warming-induced tree growth may help offset increasing disturbance across the Canadian boreal forest. Proceedings of the National Academy of Sciences, 120(2), e2212780120. https://doi.org/10.1073/pnas.2212780120

Wang, J. A., Baccini, A., Farina, M., Randerson, J. T., & Friedl, M. A. (2021). Disturbance suppresses the aboveground carbon sink in North American boreal forests. Nature Climate Change, 11(5), 435–441. https://doi.org/10.1038/s41558-021-01027-4

*AUTHOR COMMENTS: We have added Figures B7, B8, and B9, which show vegetation and soil C pools, as well as a description of these in Lines 379-381. Briefly, both versions of CLASSIC simulate similar increases in vegetation C but CLASSIC with dynamic land cover simulates a smaller decrease in soil C due to increased natural vegetation area and thus soil C storage at high latitudes, which contributes to the differences in NBP between CLASSIC with prescribed vs. dynamic landcover. Unfortunately, as described above, CLASSIC underestimates high latitude fires and thus does not capture changes in vegetation and soil C pools driven by high latitude fires. To explore the impact of the transition from needleleaf to broadleaf tree at high latitudes, we have added broadleaf and needleleaf trees to Figure 6, which shows changes in broadleaf vs. needleleaf tree area as well as broadleaf vs. needleleaf tree NPP change in SSP5-8.5, which ultimately influence NBP. We added an explanation of this transition in Lines 369-371.*

5) Have the authors compared the MODIS collection 6/6.1 to MODIS collection 5 to see if there are significant differences between the two datasets, and is there a specific reason the authors chose to use collection 5 to create the bioclimatic index instead of using collection 6?

*AUTHOR COMMENTS: We used the MODIS Collection 5 dataset because it had been previously used and processed for analyses. We briefly recalculated the bioclimatic limits with the MODIS Collection 6.1 dataset and there were only minor differences (temperature indices differed by ≤ 1°C and other indices differed by < 5%). This is a helpful suggestion and in future analyses, we will use the newest dataset.*

6) How well do the dynamic models match observational trends in annual land cover during the MODIS/ ESA CCI during the overlap of the model and observational records?

*AUTHOR COMMENTS: Thank you for this suggestion. While we agree that this analysis would yield interesting results, we think that this is beyond the scope of our study, which we want to focus on future simulations. While we do use the present-day average to evaluate historical simulations (Figures 2 and 3), evaluating the annual trends would be a significant additional analysis. It would require reclassifying remote sensing products over multiple years, then parsing land use change (i.e., the conversion of natural vegetation to crop and pasture, deforestation, etc.), which is prescribed in model simulations to allow for historical and future simulations (with the LUH2-GCB2022 dataset), from the shifting ranges of different biomes driven by climate change. This would be an entire research study on its own.*

Technical corrections/comments:

L197 Please provide citations/html sources for the GLC2000 product

*AUTHOR COMMENTS: We have added this to Line 210.*

L202 - Please provide citations/html sources for the ESA CCI product and which version you are using.

*AUTHOR COMMENTS: We have added this to Line 211.*

L202 - What years are utilized for the ESA CCI land cover product when setting up simulation 2 (S2)? Are each of the years for ESA CCI land cover utilized? Same for MODIS when creating the bioclimatic index.

*AUTHOR COMMENTS: Year 2018 from the ESACCI product was used. We added a description in Lines 215-216. This is described in detail in Wang et al. 2023. Using the methodology described in Lines 211-222, a land cover product from a single year is used to create a land cover forcing over multiple years (i.e., from 1851-2020 for historical simulations and from 2015-2100 for future simulations). To derive the bioclimatic limits, we used the Collection 5 MODIS Global Land Cover Type IGBP product time-averaged between 2001-2010 to be representative of present-day land cover. We have added a description in Lines 177-180.*

L137-8 Typo, "When crop area decreases, natural vegetation area proportionally increase"

*AUTHOR COMMENTS: We have fixed this typo.*